# COUNTING GRAPH SUBSTRUCTURES WITH GRAPH NEURAL NETWORKS

**Charilaos I. Kanatsoulis**
Electrical and Systems Engineering
University of Pennsylvania
Philadelphia, PA 19104
kanac@seas.upenn.edu

**Alejandro Ribeiro**
Electrical and Systems Engineering
University of Pennsylvania
Philadelphia, PA 19104
aribeiro@seas.upenn.edu

## ABSTRACT

Graph Neural Networks (GNNs) are powerful representation learning tools that have achieved remarkable performance in various downstream tasks. However, there are still open questions regarding their ability to count and list substructures, which play a crucial role in biological and social networks. In this work, we fill this gap and characterize the representation and generalization power of GNNs in terms of their ability to produce powerful representations that count substructures. In particular, we study the message-passing operations of GNNs with random node input in a novel fashion, and show how they can produce equivariant representations that are associated with high-order statistical moments. Using these representations, we prove that GNNs can learn how to count cycles, cliques, quasi-cliques, and the number of connected components in a graph. We also provide new insights into the generalization capacity of GNNs. Our analysis is constructive and enables the design of a generic GNN architecture that shows remarkable performance in four distinct tasks: cycle detection, cycle counting, graph classification, and molecular property prediction.

## 1 INTRODUCTION

Graph Neural Networks (GNNs) are deep learning architectures that learn powerful representations of graphs and graph signals. GNNs have shown remarkable performance across various domains, such as biology and drug discovery Gainza et al. (2020); Strokach et al. (2020); Jiang et al. (2021), quantum chemistry Gilmer et al. (2017), robotics Li et al. (2020); Hadou et al. (2022), social networks and recommender systems Ying et al. (2018). Their success can be attributed to their fundamental and favorable properties, including permutation invariance-equivariance Maron et al. (2018), stability to deformations Gama et al. (2020), and transferability Ruiz et al. (2020); Levie et al. (2021).

Since GNNs perform representation learning on graph structures and graph data, a lot of research has been conducted to answer how expressive GNN representations are. A major line of research studies the ability of GNNs to perform graph isomorphism. In particular, Morris et al. (2019); Xu et al. (2019) compare the representation power of GNNs to that of the Weisfeiler-Lehman (WL) test Weisfeiler & Leman (1968), where Kanatsoulis & Ribeiro (2022) study the expressivity of GNNs from a spectral perspective. Although graph isomorphism is an important and necessary expressivity test, it is not sufficient to fully characterize the representation power of these architectures. Consequently, there has been increased interest in approaching GNN expressivity via counting substructures Arvind et al. (2020); Chen et al. (2020) or solving graph bi-connectivity Zhang et al. (2023). In this paper, we revisit the representation power of GNNs in terms of counting important substructures of the graph. Additionally, we investigate their ability to generalize and transfer structural properties to arbitrary graph families. Our work is motivated by the following research question:

**Problem Definition 1.1** *Given a graph $\mathcal{G}$ without additional features; can a message-passing GNN learn to generate permutation equivariant representations that count and list graph substructures?*

Problem 1.1 studies the ability of a message-passing GNN to produce substructure informative representations when the input $X$ is completely uninformative. This is interesting for two reasons.

First, it provides a theoretical measure of the representation power of GNNs. Second, it has a strong practical component, as counting and listing graph substructures, also known as subgraphs, motifs, and graphlets, is pivotal in a plethora of real-world tasks. In molecular chemistry for instance the bonds between atoms or functional groups, form cycles (rings) that are indicative of a molecule's properties and can be utilized to generate molecular fingerprints Morgan (1965); Alon et al. (2008); Rahman et al. (2009); O'Boyle & Sayle (2016). Cliques and quasi-cliques, on the other hand, characterize protein complexes in Protein-Protein Interaction networks and community structure in social networks Girvan & Newman (2002); Jiang et al. (2010); Fox et al. (2020).

In this work, we give an affirmative answer to the Problem 1.1 and analyze the representation power of GNNs in terms of their ability to count and list cycles, cliques, quasi-cliques, and connected components. To this end, we consider a standard GNN architecture, that performs local message-passing operations, combined with normalization layers. Our analysis employs tools from tensor algebra and stochastic processes to characterize the output of the GNN, when the input is a random vector. We show that a suitable choice of activation and normalization functions can generate equivariant node representations, which capture the statistical moments of the GNN output distribution. Additionally, we derive a deterministic, closed-form expression for the GNN output and provide a rigorous proof that accurately lists all the triangles, and counts cycles ranging from size 3 to 8, 4-node cliques, quasi-cliques, as well as connected components within the graph.

Our results also offer novel insights into the generalization properties of message-passing GNNs. Specifically, we demonstrate that a GNN can learn to count important substructures, not only within the family of graphs observed during training but also for any arbitrary graph. This analysis is constructive and enables the development of a generic architecture that exhibits success across various tasks. Extensive numerical tests validate the effectiveness of the proposed approach across four distinct tasks: cycle detection, cycle counting, graph classification, and molecular property prediction. Our contribution is summarized as follows:

(C1) We characterize the representation and generalization power of message-passing GNNs by assessing their ability to count and list important graph substructures.

(C2) We propose a novel analysis of GNNs with random node inputs that unlocks the true potential of message-passing, and enables them to generate powerful equivariant representations that count or list cycles, quasi-cliques, and connected components within a graph.

(C3) We prove that the substructure counting ability of GNNs is generalizable and can transfer to arbitrary graphs that were not observed during training.

(C4) Our constructive analysis enables a generic architecture that is effective across different tasks.

**Related work:** There are three main lines of work that are closely related to our paper. The first by Xu et al. (2019); Morris et al. (2019); Arvind et al. (2020); Chen et al. (2020) establishes that message-passing GNNs with degree-related inputs are, at most, as powerful as the WL test, thus limiting their ability to count simple subgraphs as triangles. The second line of work Abboud et al. (2021); Sato et al. (2021) addresses these limitations by employing random features as input to the GNN. This results in enhanced function approximation properties, but at the expense of permutation equivariance, which is fundamental in graph learning. Finally, Tahmasebi et al. (2020); You et al. (2021); Bouritsas et al. (2022); Barceló et al. (2021); Bevilacqua et al. (2021); Zhang & Li (2021); Zhao et al. (2021) directly augment GNNs with structural information to enhance their representation power. A detailed discussion of prior work with additional references can be found in Appendix A.

## 2 MESSAGE-PASSING GNNS

To begin our discussion consider a graph $\mathcal{G} := (\mathcal{V}, \mathcal{E})$, with a set of vertices $\mathcal{V} = \{1, \ldots, N\}$, a set of edges $\mathcal{E} = \{(v, u)\}$, and an adjacency matrix $\boldsymbol{S} \in \{0, 1\}^{N \times N}$. The vertices (nodes) of the graph are often associated with graph signals $\boldsymbol{x}_v \in \mathbb{R}^D$ with $D$ features, also known as node attributes.

In this paper, we study standard GNNs that perform local, message-passing operations. A message-passing GNN is a cascade of layers and is usually defined by the following recursive formula:

$$\boldsymbol{x}_v^{(l)} = g^{(l-1)}\left(\boldsymbol{x}_v^{(l-1)}, f^{(l-1)}\left(\left\{\boldsymbol{x}_u^{(l-1)} : u \in \mathcal{N}(v)\right\}\right)\right), \tag{1}$$

where $\mathcal{N}(v)$ is the neighborhood of vertex $v$, i.e., $u \in \mathcal{N}(v)$ if and only if $(u, v) \in \mathcal{E}$. The function $f^{(l)}$ aggregates information from the multiset of neighboring signals, whereas $g^{(l)}$ combines the signal of each vertex with the aggregated one from the neighboring vertices. Common choices for $f^{(l)}$, $g^{(l)}$ are the multi-layer perceptron (MLP), the linear function, and the summation function. When $f^{(l)}$ is the summation function, and $g^{(l)}$ is the multivariate linear function for $l = 1, \ldots K - 1$ and the MLP followed by normalization for $l = K$, the output of the $K$-th layer is equivalent to the following expression (as shown in Appendix B):

$$\boldsymbol{Y} = \rho \left[ \boldsymbol{X}^{(K)} \right] = \rho \left[ \sigma \left( \sum_{k=0}^{K} \boldsymbol{S}^k \boldsymbol{X} \boldsymbol{H}_k \right) \right], \tag{2}$$

where $\boldsymbol{X} = \boldsymbol{X}^{(0)} \in \mathbb{R}^{N \times D}$ represents the signals of all vertices, $\boldsymbol{H}_k \in \mathbb{R}^{D \times F}$ are the trainable parameters, $\sigma$ is a point-wise nonlinear activation function, and $\rho$ is a normalization function, e.g., batch normalization or softmax. Note that the $K$-th layer is not necessarily the final one, i.e., $\boldsymbol{Y}$ can be used as an input to other layers defined in 1. Unless noted otherwise the term GNN will refer to local, message-passing GNNs.

## 3 STUDYING GNNs WITH RANDOM INPUT

Throughout the rest of this paper, we will utilize tensor algebra, specifically the tensor mode product, the canonical polyadic decomposition (CPD) model, and the Tucker model. For a brief introduction, please refer to Appendix C, and for a more comprehensive understanding of tensors, we recommend consulting Sidiropoulos et al. (2017); Kolda & Bader (2009).

The goal of our paper is to answer the research question in 1.1 and characterize the expressive power of message-passing GNNs in terms of counting important substructures of the graph. To do that we study Equation 2, when the node input is a random vector $\boldsymbol{x} \in \mathbb{R}^N$. Our work focuses on the ability of message-passing GNN operations to generate expressive features, therefore the input $\boldsymbol{x}$ should be completely uninformative, i.e., structure and identity agnostic. To ensure that $\boldsymbol{x} = [x_1, x_2, \ldots, x_N]^T$ is agnostic with respect to the structure of the graph, we assume that $x_i, \ i = 1, \ldots, N$ are independent random variables. To ensure that $x_i, \ i = 1, \ldots, N$ are identity agnostic, we assume that they are identically distributed and satisfy $\mathbb{E}[x_i] = 0$, $\mathbb{E}[x_i^p] = 1$, for $i = 1, \ldots, N$, and $p \geq 2 \in \mathbb{Z}$. Overall, the elements of $\boldsymbol{x}$ are independent and identically distributed (i.i.d.), with joint characteristic function $\phi_{\boldsymbol{x}}(t_1, \ldots, t_N) = \prod_{i=1}^{N} \left( e^{j t_i} - j t_i \right)$, and the moments of the joint distribution of $\boldsymbol{x}$ satisfy:

$$\mathbb{E}[\boldsymbol{x}] = \boldsymbol{0}, \quad \mathbb{E}[\boldsymbol{x} \boldsymbol{x}^T] = \boldsymbol{I}, \quad \mathbb{E}[\boldsymbol{x} \circ \boldsymbol{x} \circ \boldsymbol{x}] = \underline{\boldsymbol{I}}, \quad \mathbb{E}[\boldsymbol{x} \circ \boldsymbol{x} \circ \cdots \circ \boldsymbol{x}] = \underline{\boldsymbol{I}}, \tag{3}$$

where $\circ$ is the outer product and $\underline{\boldsymbol{I}}$ denotes a super-diagonal tensor with all-one values, i.e., $\underline{\boldsymbol{I}}[i_1, i_2, \ldots, i_q] = 1$ if $i_1 = i_2 = \cdots = i_q$ and $\underline{\boldsymbol{I}}[i_1, i_2, \ldots, i_q] = 0$ otherwise. Note that $\mathbb{E}[\boldsymbol{x}] = \boldsymbol{0}$ is an 1-dimensional tensor (vector), $\mathbb{E}[\boldsymbol{x} \boldsymbol{x}^T] = \mathbb{E}[\boldsymbol{x} \circ \boldsymbol{x}] = \boldsymbol{I}$ is a 2-dimensional tensor (matrix), and $\mathbb{E}[\boldsymbol{x} \circ \boldsymbol{x} \circ \boldsymbol{x}] = \underline{\boldsymbol{I}}$, $\mathbb{E}[\boldsymbol{x} \circ \boldsymbol{x} \circ \cdots \circ \boldsymbol{x}] = \underline{\boldsymbol{I}}$ are 3- and multi-dimensional tensors respectively.

We process the input vector $\boldsymbol{x}$ using 2, i.e., $\boldsymbol{Y} = \rho \left[ \sigma \left( \sum_{k=0}^{K} \boldsymbol{S}^k \boldsymbol{x} \boldsymbol{h}_k^T \right) \right]$, where $\boldsymbol{h}_k \in \mathbb{R}^F$ represents a set of $F$ graph filters of the form:

$$\boldsymbol{z} = \sum_{k=0}^{K} h_k \boldsymbol{S}^k \boldsymbol{x} = \boldsymbol{H}(\boldsymbol{S}) \boldsymbol{x}, \quad \boldsymbol{y} = \rho[\sigma(\boldsymbol{z})], \tag{4}$$

where $\boldsymbol{H}(\boldsymbol{S}) = \sum_{k=0}^{K} h_k \boldsymbol{S}^k$. To produce equivariant, deterministic node features we choose $\sigma, \rho$ such that they can instantiate moments of the distribution of $\boldsymbol{z}$. As we see in the next section, the moments of $\boldsymbol{z}$ can count some very important graph substructures.

## 4 COUNTING GRAPH SUBSTRUCTURES WITH GNNs

In this section, show that appropriate choices of $\sigma, \rho$ in 4 generate equivariant node features that are associated with the moments of the distribution of $\boldsymbol{z}$. We then analyze these moments and prove that they count cycles, quasi-cliques, and connected components in a graph. The proofs can be found in Appendices D to I. Results for counting cycles at the node level can be found in Appendix K.

### 4.1 SECOND ORDER MOMENT

First, we study 4 when $\sigma$ is the elementwise square function and $\rho$ is the expectation operator. This architecture computes deterministic equivariant node features that measure the variance of $z$, i.e., $y = \rho\left[\sigma\left(z\right)\right] = \mathbb{E}\left[z^2\right]$. Note that $\mathbb{E}\left[z\right] = 0$, since $\mathbb{E}\left[x\right] = 0$ and $x, z$ are linearly related. To analyze the output $y$ we instantiate the covariance of $z$ that takes the form:

$$\mathbb{E}\left[zz^T\right] = \mathbb{E}\left[\sum_{k=0}^{K} h_k S^k x x^T \sum_{m=0}^{K} h_m S^{m^T}\right] = \sum_{k=0}^{K} h_k S^k \mathbb{E}\left[xx^T\right] \sum_{m=0}^{K} h_m S^m = H\left(S\right) H\left(S\right),$$

where the last equality comes from the fact that $H\left(S\right) = \sum_{k=0}^{K} h_k S^k$ and $\mathbb{E}\left[xx^T\right] = I$. Then:

$$y = \rho\left[\sigma\left(z\right)\right] = \mathbb{E}\left[z^2\right] = \operatorname{diag}\left(\mathbb{E}\left[zz^T\right]\right) = \left(H\left(S\right) \odot H\left(S\right)\right)\mathbf{1}, \tag{5}$$

where $\operatorname{diag}(A) \in \mathbb{R}^N$ is the vector with the diagonal of $A \in \mathbb{R}^{N \times N}$, $\odot$ denotes the Hadamard (elementwise) product operation, and the last equality comes from the following property, $\operatorname{diag}\left(AB\right) = \left(A \odot B\right)\mathbf{1}$, for square matrices $A, B$. Expanding 5 yields the following expression:

$$y = \left(\sum_{k=0}^{K} h_k S^k \odot \sum_{m=0}^{K} h_m S^m\right)\mathbf{1} = \sum_{k=0}^{K}\sum_{m=0}^{K} h_k h_m \left(S^k \odot S^m\right)\mathbf{1} = \sum_{k,m} h_{k,m}\left(S^k \odot S^m\right)\mathbf{1}.$$

$$\tag{6}$$

We leverage the expression in 6 to prove the following theorems.

**Theorem 4.1 (Number of connected components)** *Given a set of graphs $\{\mathcal{G}_i\}_{i=1}^{M}$, there exists a GNN defined in 1 or 2 that counts the number of connected components of all $\{\mathcal{G}_i\}_{i=1}^{M}$.*

**Theorem 4.2 (Cycles)** *There exists a GNN defined in 1 or 2 that counts the number of cycles with 3 to 5 nodes, of any graph.*

Note that in Theorem 4.1 we prove that a GNN is able to learn how to count the connected components of graphs that were observed during training. Theorem 4.2, one the other hand, proves that a GNN is able to learn how to count the 3-,4-,5-node cycles of any graph. As a result, Theorem 4.2 is stronger and also provides novel insights into the generalization ability of GNNs.

### 4.2 THIRD-ORDER MOMENT

Next, we study 4 when $\sigma$ is the elementwise cubic function and $\rho$ is the expectation operator. This architecture computes deterministic equivariant node features that measure the skewness of $z$, i.e., $y_1 = \rho\left[\sigma\left(z\right)\right] = \mathbb{E}\left[z^3\right]$. To analyze the output $y$ we instantiate the third-order moment of $z$, which is a super-symmetric third-order tensor and is defined as follows:

$$\mathbb{E}\left[z \circ z \circ z\right] = \mathbb{E}\left[x \circ x \circ x\right] \times_1 H\left(S\right) \times_2 H\left(S\right) \times_3 H\left(S\right), \tag{7}$$

where "$\times_1, \times_2, \times_3$" denote the tensor mode products and multiply each column, row, and fiber of $\mathbb{E}\left[x \circ x \circ x\right]$ with $H\left(S\right)$ respectively. Since $\mathbb{E}\left[x \circ x \circ x\right] = \underline{I}$, 7 takes the form:

$$\mathbb{E}\left[z \circ z \circ z\right] = \underline{I} \times_1 H\left(S\right) \times_2 H\left(S\right) \times_3 H\left(S\right) = [\![H\left(S\right), H\left(S\right), H\left(S\right)]\!], \tag{8}$$

where $[\![H\left(S\right), H\left(S\right), H\left(S\right)]\!] = \sum_{n=1}^{N} H\left(S\right)[:, n] \circ H\left(S\right)[:, n] \circ H\left(S\right)[:, n]$ denotes the CPD model of $\mathbb{E}\left[z \circ z \circ z\right]$ with factor matrices $H\left(S\right)$. Then $y_1$ takes the form:

$$y_1 = \rho\left[\sigma\left(z\right)\right] = \mathbb{E}\left[z^3\right] = \operatorname{superdiag}\left(\mathbb{E}\left[z \circ z \circ z\right]\right) = \left(H\left(S\right) \odot H\left(S\right) \odot H\left(S\right)\right)\mathbf{1}, \tag{9}$$

where $\operatorname{superdiag}\left(\cdot\right) \in \mathbb{R}^N$ is the superdiagonal of the tensor and the last equality comes from the definition of the CPD model. Interestingly, 9 can be cast as follows:

$$y_1 = \left(\sum_{k=0}^{K} h_k S^k \odot \sum_{l=0}^{K} h_l S^l \odot \sum_{m=0}^{K} h_m S^m\right)\mathbf{1} = \sum_{k,l,m=0}^{K} h_{k,l,m}\left(S^k \odot S^l \odot S^m\right)\mathbf{1}. \tag{10}$$

Using the expression in 10 we prove in the Appendix the following theorems:

**Theorem 4.3 (6-node Cycles)** *There exists a GNN defined in 1 or 2 that counts the number of 6-node cycles of any graph.*

**Theorem 4.4 (Quasi-cliques)** *There exists a GNN defined in 1 or 2 that counts the number of 4-node and 5-node quasi-cliques (chordal cycles) of any graph.*

Theorems 4.3 and 4.4 indicate that $\boldsymbol{y}_1 = \mathbb{E}\left[\boldsymbol{z}^3\right]$ is crucial in counting 6-node cycles and 4,5-node quasi-cliques. We can also linearly derive additional equivariant features, from the third moment tensor $\mathbb{E}\left[\boldsymbol{z} \circ \boldsymbol{z} \circ \boldsymbol{z}\right]$, that produce rich information about the graph. In particular, we can compute the following equivariant feature vector, which is essential in our proofs for 7-node cycles.

$$\boldsymbol{y}_2 = \mathrm{diag}\left(\mathbb{E}\left[\boldsymbol{z} \circ \boldsymbol{z} \circ \boldsymbol{z}\right] \times_3 \mathbf{1}^T\right) = \mathrm{diag}\left(\mathbb{E}\left[\boldsymbol{z} \circ \boldsymbol{z} \circ \mathbf{1}^T \boldsymbol{z}\right]\right) = \mathbb{E}\left[\boldsymbol{z}^2 \left(\mathbf{1}^T \boldsymbol{z}\right)\right]. \qquad (11)$$

The expression in 11 can be derived from 4 when $\sigma$ is the elementwise square function, and $\rho\left(\cdot\right) = \mathbb{E}\left[\left(\cdot\right) \odot \left(\mathbf{1}^T \boldsymbol{z}\right)\right]$. In that case, $\rho$ defines a normalization layer that multiplies each datum in $\boldsymbol{z}^2$ by $\mathbf{1}^T \boldsymbol{z}$ and measures the average of all data. After some algebraic manipulations that can be found in the Appendix the expression in 11 takes the form:

$$\boldsymbol{y}_2 = \left(\sum_{k=0}^{K} h_k \boldsymbol{S}^k \odot \sum_{l=0}^{K} h_l \boldsymbol{S}^l\right) \sum_{m=0}^{K} h_m \boldsymbol{S}^m \mathbf{1} = \sum_{k,l,m=0}^{K} h_{k,l,m} \left(\boldsymbol{S}^k \odot \boldsymbol{S}^l\right) \boldsymbol{S}^m \mathbf{1}. \qquad (12)$$

### 4.3 FOURTH-ORDER MOMENT

Following the idea of the previous subsections, if $\sigma$ is the elementwise quartic function and $\rho$ is the expectation, we can compute deterministic equivariant node features that measure the kurtosis of $\boldsymbol{z}$.

$$\boldsymbol{y} = \mathbb{E}\left[\boldsymbol{z}^4\right] = \sum_{k,l,m,n=0}^{K} h_{k,l,m,n} \left(\boldsymbol{S}^k \odot \boldsymbol{S}^l \odot \boldsymbol{S}^m \odot \boldsymbol{S}^n\right) \mathbf{1}. \qquad (13)$$

Using 6 , 12, 13 we can derive the following theorem:

**Theorem 4.5 (7-node Cycles)** *There exists a GNN defined in 1 or 2 that counts the number of 7-node cycles of any graph.*

**Remark 4.6** *Theorems 4.2, 4.3, 4.4, and 4.5 prove the ability of a GNN to learn how to count the substructures of any graph. This brings new insights into the generalization ability of GNNs.*

## 5 HIGHER-ORDER MOMENTS AND MULTIPLE-LAYER ANALYSIS

In this section, we generalize our analysis and study a multi-layer message-passing GNN that computes higher-order moments. To this end, we present the following proposition.

**Proposition 5.1** *Consider a message-passing GNN with $L$ layers, where the output of each layer is defined by 2. Let the input to the GNN be a random vector with moments as in 3, and $\sigma$ be the elementwise $m$−th power, i.e., $\sigma_l\left(\cdot\right) = \left(\cdot\right)^{m_l}$ for $l = 1, \dots, Q$. Also let $\rho\left(\cdot\right) = \mathbb{1}\left(\cdot\right)$ be the identity function in the first $L - 1$ layers, and $\rho\left(\cdot\right) = \mathbb{E}\left[\left(\cdot\right) \odot \left(\mathbf{1}^T \boldsymbol{z}\right)^{i_m}\right]$ only in the $L$−th layer. Then the output $\boldsymbol{Y} = \left[\boldsymbol{y}_1, \dots, \boldsymbol{y}_F\right] \in \mathbb{R}^{N \times F_L}$ of this GNN, has a closed-form expression, defined by 14.*

$$\boldsymbol{y}_f = \sum_{i_1,\dots,i_n=0}^{K} h_{i_1,\dots,i_n} \underbrace{\left(\boldsymbol{S}^{i_1} \odot \cdots \odot \boldsymbol{S}^{i_k}\right) \cdots \left(\boldsymbol{S}^{i_l} \odot \cdots \odot \boldsymbol{S}^{i_m}\right)}_{L\ times} \left(\boldsymbol{S}^{i_{m+1}} \mathbf{1} \odot \cdots \odot \boldsymbol{S}^{i_n} \mathbf{1}\right) \quad (14)$$

The number of multiplications $L$ in 14 is equal to the number of layers, the term $\left(\boldsymbol{S}^{i_1} \odot \cdots \odot \boldsymbol{S}^{i_k}\right)$ reflects the effect of the activation $\sigma(\cdot)$, and $\left(\boldsymbol{S}^{i_{m+1}} \mathbf{1} \odot \cdots \odot \boldsymbol{S}^{i_n} \mathbf{1}\right)$ is affected by the choice of normalization $\rho$ in the final layer. A schematic illustration of the considered architecture is presented in Fig. 1. The module 1a consists of proper message-passing operations, whereas the module 1b consists of Hadamard products between adjacency powers and sparse matrix-vector multiplications. The computational complexity of our approach is analyzed in Appendix L.

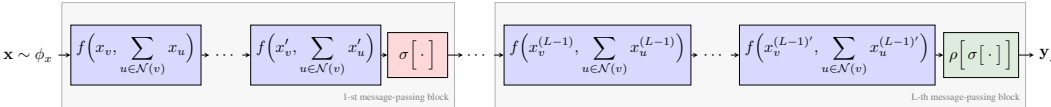

(a) Message-passing module with random input

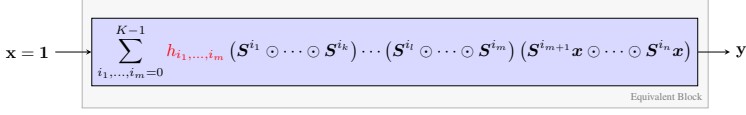

(b) Equivalent module

Figure 1: `Moment-GNN` modules

**Remark 5.1** *Modules 1a and 1b are equivalent and can be used interchangeably. Furthermore, Layer L in Proposition 5.1 or Fig. 1 need not need to be the final layer of the GNN architecture, can be followed by classical message-passing layers as defined in 1.*

The previous analysis can also be used to characterize the expressive power of the GNNs defined in 1 or 2 with respect to that of the folklore-Weisfeiler-Lehman (FWL) test Cai et al. (1992); Morris et al. (2019); Huang & Villar (2021).

**Proposition 5.2 (Expressive Power)** *A GNN defined by Proposition 5.1 followed by layers described in 1 is strictly more powerful than the 1-FWL test, when $f$, $g$ are injective functions.*

# 6    LISTING AND COUNTING STRUCTURES BEYOND 2-FWL

In this section, we show how a message-passing GNN can generate equivariant features that list all the triangles of a graph, count the number of $4-$node cliques, $8-$node cycles, and therefore break the expressivity limits of the 2-FWL test. In particular, we consider again the GNN described in 2, which we study with an uninformative input as described in Section 3, to derive the following theorem.

**Theorem 6.1 (Listing Triangles)** *There exists a message-passing GNN defined in 15:*

$$\underline{\boldsymbol{T}} = ReLU\left(\mathbb{E}\left[\boldsymbol{z} \circ \boldsymbol{z} \circ \boldsymbol{z}\right]\right) \in \{0,1\}^{N \times N \times N}, \quad \boldsymbol{z} = \sum_{k=0}^{K} h_k \boldsymbol{S}^k \boldsymbol{x}, \tag{15}$$

*with an uninformative input $\boldsymbol{x} \in \mathbb{R}^N$ as described in Section 3 such that $\underline{\boldsymbol{T}}\left[i,j,k\right] = 1$ if vertices $i, j, k$ form a triangle and $\underline{\boldsymbol{T}}\left[i,j,k\right] = 0$ otherwise.*

In Theorem 6.1 we start with i.i.d. random node inputs, as in the previous analysis, and perform linear message-passing operations to compute informative node representations $\boldsymbol{z}$. Instead of applying a pointwise nonlinearity to $\boldsymbol{z}$, we compute the three-way outer product $\boldsymbol{z} \circ \boldsymbol{z} \circ \boldsymbol{z}$, and process it via the expectation and ReLU. This enables the computation of tensor $\underline{\boldsymbol{T}}$, which encodes relations of node triplets and can list all the triangles in the graph. After learning tensor $\underline{\boldsymbol{T}}$ we can use it to generate a random vector $\boldsymbol{x}_t$ with the following statistical moments:

$$\mathbb{E}\left[\boldsymbol{x}_t\right] = \boldsymbol{0}, \quad \mathbb{E}\left[\boldsymbol{x}_t \circ \boldsymbol{x}_t \circ \boldsymbol{x}_t\right] = \underline{\boldsymbol{T}}. \tag{16}$$

We can then perform a similar analysis as the one in Section 4 and derive the following theorems.

**Theorem 6.2 (4-node cliques)** *There exists a GNN defined in 1 or 2 with random input $\boldsymbol{x}_t$ defined in 16, that counts the number of 4-node cliques of any graph.*

**Theorem 6.3 (8-node Cycles)** *There exists a GNN defined in 1 or 2 with random input $\boldsymbol{x}_t$ defined in 16, that counts the number of 8-node cycles of any graph.*

The details of the analysis can be found in Appendix J. Theorems 6.2, 6.3 also prove that:

**Proposition 6.1 (Expressive Power)** *There exist substructure families with $\mathcal{O}\left(1\right)$, such that 2-FWL is no stronger than the GNN described in Theorems 6.2, 6.3.*

Table 1: Cycle detection accuracy for different GNN models

| Cycle length | 4 | | | | 6 | | | | 8 | | | |
|---|---|---|---|---|---|---|---|---|---|---|---|---|
| Graph size | 12 | 20 | 28 | 36 | 20 | 31 | 42 | 56 | 28 | 50 | 66 | 72 |
| Neural-MP | 98.5 | 93.2 | 91.8 | 86.7 | 98.7 | 95.5 | 92.9 | 88.0 | 98.0 | 96.3 | 92.5 | 89.1 |
| GIN | 98.3 | 97.1 | 95.0 | 93.0 | 99.5 | 97.2 | 95.1 | 92.7 | 98.5 | 98.8 | 90.8 | 92.5 |
| GIN + degree | 99.3 | 98.2 | 97.3 | 96.7 | 99.2 | 97.1 | 97.1 | 94.5 | 99.3 | 98.7 | | 95.4 |
| GIN + rand id | 99.0 | 96.2 | 94.9 | 88.3 | 99.0 | 97.8 | 95.1 | 96.1 | 98.6 | 98.0 | 97.2 | 95.3 |
| RP | 100 | 99.9 | 99.7 | 97.7 | 99.0 | 97.4 | 92.1 | 84.1 | 99.2 | 97.1 | 92.8 | 80 |
| PPGN | 100 | 100 | 100 | 99.8 | 98.3 | 99.4 | 93.8 | 87.1 | 99.9 | 98.7 | 84.4 | 76.5 |
| Ring-GNN | 100 | 99.9 | 99.9 | 99.9 | 100 | 100 | 100 | 100 | 99.1 | 99.8 | 74.4 | 71.4 |
| SMP | 100 | 100 | 100 | 100 | 100 | 100 | 100 | 100 | 100 | 100 | 100 | 99.9 |
| Moment-GNN | 100 | 99.9 | 100 | 99.9 | 100 | 100 | 100 | 100 | 100 | 100 | 100 | 100 |

Table 2: Cycle detection for in- and out-of-distribution graphs

| Setting | In-distribution | | | Out-of-distribution | | |
|---|---|---|---|---|---|---|
| Cycle length | 4 | 6 | 8 | 4 | 6 | 8 |
| Graph size | 20 | 31 | 50 | 36 | 56 | 72 |
| GIN | 93.9 | 99.7 | 98.8 | 81.1 | 85.8 | 88.8 |
| PPGN | 99.9 | 99.5 | 98.7 | 50.0 | 50.0 | 50.0 |
| Ring-GNN | 100 | 100 | 99.9 | 50.0 | 50.0 | o.o.m. |
| SMP | 100 | 99.8 | 99.5 | 99.8 | 87.8 | 79.5 |
| Moment-GNN | 100 | 100 | 99.1 | **100** | **100** | **100** |

# 7 EXPERIMENTS

## 7.1 ARCHITECTURE

To implement the proposed approach, which we denote as `Moment-GNN`[1], we use the equivalent model, shown in Fig. 1b and Equation 14. The architecture consists of a single `Moment-GNN` layer followed by $2 - 6$ message-passing GIN layers Xu et al. (2019), defined by 1, where $f$ is the summation function and $g$ is an MLP. The `Moment-GNN` uses the closed-form expression in 14 (or more specifically in 65) to compute the individual moments $\mathbb{E}\left[\boldsymbol{z}^m\right]$, for $m = 2, 3, 4, 5$. We use a set of $F_m$ filters for the $m$-th moment, which outputs a feature representation $\boldsymbol{Y} \in \mathbb{R}^{N \times (F_2 + F_3 + F_4 + F_5)}$ the is used as an input to the GIN layers. The maximum $m$ and the number of layers are hyperparameters.

## 7.2 CYCLE DETECTION

In the first set of experiments, we test the ability of the proposed approach to detect cycles of lengths 4, 6, and 8 in graphs of different sizes, which is a binary graph classification task. The baselines used for comparison are: Neural message-passing network (`Neural-MP`) Gilmer et al. (2017), `GIN` Xu et al. (2019), relational pooling GNN (`RP`) Murphy et al. (2019), provably powerful GNN (`PPGN`) Maron et al. (2019), `Ring-GNN` Chen et al. (2019), and structural message-passing GNN (`SMP`) Vignac et al. (2020). Note that `RP`, `PPGN`, `Ring-GNN`, and `SMP` have significantly higher computational and memory complexity compared to `Neural-MP`, `GIN`, and `Moment-GNN`. We also consider two variants of `GIN`, namely `GIN + degree` and `GIN + rand id`. The input of `GIN + degree` is the one-hot encoding of the degree of each node, and the input of `GIN + rand id` is a random vector that works as a unique identifier of each node Abboud et al. (2021); Sato et al. (2021). The latter increases the expressiveness of `GIN` at the expense of permutation equivariance.

We use the dataset and procedure described in Vignac et al. (2020)[2]. Each model is trained with 10,000 graphs for each cycle length. Testing is performed on a separate set of 10,000 graphs. The classification accuracy of the competing GNNs for different cycle lengths and graph sizes is presented in Table 1. The results for the baselines are taken from Vignac et al. (2020).

---

[1]https://github.com/MomentGNN
[2]https://github.com/cvignac/SMP

Table 3: Moment-GNN Performance in ZINC

(a) Regression MAE for counting cycles in ZINC

| Pentagon | | Hexagon | |
|---|---|---|---|
| Training | Testing | Training | Testing |
| 0.0012 | 0.0021 | 0.0057 | 0.0059 |

(b) Classification accuracy cycle detection in ZINC

| Nonagon | | Decagon | |
|---|---|---|---|
| Training | Testing | Training | Testing |
| 100 | 99.8 | 99.9 | 99.4 |

We observe that `Moment-GNN` and `SMP` manage to perfectly detect the cycles of the graphs, whereas `Neural-MP`, `RP`, `PPGN`, `Ring-GNN` show weaker performance in larger graphs. It is notable that `Moment-GNN` is a message-passing GNN with lower complexity than `RP`, `PPGN`, `RING-GNN`, yet it yields enhanced performance. Furthermore, `Moment-GNN` markedly outperforms `GIN + rand id`, despite the initial resemblance in their utilization of random input for feeding a message-passing GNN. This can be attributed to the fact that `GIN + rand id` generates random realizations that serve as distinct identifiers, thus negating permutation equivariance. On the contrary, `Moment-GNN` employs a stochastic approach of the random input and maintains permutation equivariance, which is a crucial property for this task.

In the second set of experiments, we examine the generalization and transferability of the different GNN models. In particular, we train the GNN architectures to detect the cycles of small graphs and test detection performance with larger graphs. The classification accuracy for in-distribution and out-of-distribution graphs are presented in Table 2 (o.o.m stands for out of memory). We observe that `Moment-GNN` is able to perfectly detect the cycles in all settings. In particular, it achieves 20% improved accuracy compared to `SMP` in detecting out-of-distribution cycles of length 8 and 12% improvement in detecting out-of-distribution cycles of length 6. It is also 50% better than `PPGN` and `Ring-GNN`. The result becomes even more impressive if we consider that `SMP`, `PPGN` and `Ring-GNN` have higher computational and memory complexity compared to `Moment-GNN`.

### 7.3 CYCLE COUNTING

We further test the ability of `Moment-GNN` to count and detect cycles. To this end, we consider the ZINC dataset, which consists of $12,000$ molecular graphs of different chemical compounds Irwin et al. (2012); Dwivedi et al. (2023). The cycle statistics of ZINC graphs are presented in Fig. 2. We observe that the prevailing substructure is the hexagon (cyclohexane), which is expected since cyclohexanes are the most common rings in molecular conformations. There are also several pentagons and some single nonagons and decagons. The number of the remaining cycles is insignificant and it is unlikely that a learning method will be able to count them. Following these observations we train `Moment-GNN` for 4 different tasks: i) count the number of pentagons, ii) count the number of hexagons, iii) detect a nonagon, iv) detect a decagon. The first two are formulated as regression tasks with $L_1$ loss, and the last two as binary classification with logistic loss. The ground-truth number of cycles was computed using the subgraph isomorphism algorithm VF2 Cordella et al. (2004). For the detection tasks, we train and test with a subset of the available graphs to ensure balanced classes.

The training and testing results for counting pentagons and hexagons are presented in Table 3a. We observe that `Moment-GNN` is able to count these cycles with very high accuracy, as the mean absolute error (MAE) for both tasks is in the order of $10^{-3}$. The results for nonagon and decagon detection are presented in Table 3b. `Moment-GNN` is able to achieve almost perfect training and testing classification accuracy for both cycle types, even though our theory does not guarantee that the proposed framework will be able to do so. Baseline comparisons can be found in Appendix M.

### 7.4 LOGP PREDICTION

Next, we consider the task of predicting the penalized water-octanol partition coefficient-logP for molecules in the ZINC dataset. We use 10,000 of them for training, 1,000 for validation and 1,000 for testing. We use a 5-layer message-passing GNN, where the first layer is a Moment-GNN and the remaining are traditional message-passing layers with skip connections followed by batch-normalization layers. We use 2 different implementations; one that does not process edge features and one that does. The feature dimension is set to 128. To assess the performance of our approach we use the MAE between the estimated and the ground-truth logP of the testing set. We perform 10

Table 4: logP Prediction in ZINC

| GNN Model | MAE | MAE (EF) |
|---|---|---|
| GraphSage | $0.410 \pm 0.005$ | − |
| GAT | $0.463 \pm 0.002$ | − |
| MoNet | $0.407 \pm 0.007$ | − |
| GatedGCN | $0.422 \pm 0.006$ | $0.363 \pm 0.009$ |
| PNA | $0.320 \pm 0.032$ | $0.188 \pm 0.004$ |
| DGN | $0.219 \pm 0.0102$ | $0.168 \pm 0.003$ |
| GNNML | $0.161 \pm 0.006$ | − |
| HIMP | − | $0.151 \pm 0.006$ |
| SMP | $0.219\pm$ | $0.138\pm$ |
| GIN + rand id | $0.322 \pm 0.026$ | $0.279 \pm 0.023$ |
| GCN | $0.469 \pm 0.002$ | − |
| GIN | $0.254 \pm 0.014$ | $0.209 \pm 0.018$ |
| GSN with cycles | $\mathbf{0.140 \pm 0.006}$ | $0.115 \pm 0.012$ |
| **Moment-GNN** | $\mathbf{0.140 \pm 0.004}$ | $\mathbf{0.110 \pm 0.005}$ |

Table 5: Graph classification on REDDIT

| Method | REDDIT-B | REDDIT-M |
|---|---|---|
| WLkernel | $81.0 \pm 3.1$ | $52.5 \pm 2.1$ |
| - | − | − |
| - | − | − |
| - | − | − |
| WEGL | $\mathbf{92.9 \pm 1.9}$ | $55.4 \pm 1.6$ |
| AWL | $87.9 \pm 2.5$ | $50.5 \pm 1.9$ |
| DGK | $78.0 \pm 0.4$ | $41.3 \pm 0.2$ |
| PATCHYSAN | $86.3 \pm 1.6$ | $49.1 \pm 0.7$ |
| RetGK | $90.8 \pm 0.2$ | $54.2 \pm 0.3$ |
| GIN + rand id | $91.8 \pm 1.6$ | $57.0 \pm 2.1$ |
| GCN | $50.0 \pm 0.0$ | $20.0 \pm 0.0$ |
| GIN | $91.8 \pm 1.1$ | $56.9 \pm 2.1$ |
| GSN with cliques | $91.1 \pm 1.8$ | $56.2 \pm 1.8$ |
| **Moment-GNN** | $92.0 \pm 1.8$ | $\mathbf{58.1 \pm 1.6}$ |

different runs and report the mean and standard deviation of the MAE of the model that showed the best validation accuracy. We compare against various state-of-art baselines and report the results in Table 4. We observe that `Moment-GNN` achieves state-of-the-art performance in predicting logP for molecular graphs, whereas GSN with cycle features is the second best. The performance of `Moment-GNN` is achieved by message-passing operations without additional knowledge. This is contrary to GSN which precomputes substructure counts and fine tunes according to them.

## 7.5 GRAPH CLASSIFICATION

We also test the performance of Moment-GNN in classifying the graphs of REDDIT-B (2000 graphs, 2 classes, 429.6 Avg# nodes) and REDDIT-M (5000 graphs, 5 classes, 508.5 Avg# nodes) datasets. Note that the majority of powerful GNNs cannot operate on these datasets due to their large size. For example, GSN is only able to compute a small number of structural features. To train the GNN models, the graphs are divided into 50-50 training testing splits and 10 different folds. Once again, we use a 5-layer GNN as before but without skip-connections. The feature dimension is set to 64. Table 5 presents the mean and standard deviation of the classification accuracy over 10 folds. We report the epoch that provided the best results among 350 epochs, which is the standard for this dataset.

From Table 6 we see that `Moment-GNN` achieves state-of-the-art performance for one more task. Note that `Moment-GNN` is a generic architecture and there is no need to predefine the structure type (e.g., cycle vs clique, induced vs non-induced) required for a specific task. `Moment-GNN` will decide the appropriate substructure type according to the data and the task.

In summary, our findings highlight the remarkable performance of `Moment-GNN` across 4 different tasks. Tables 1, 3a show that `Moment-GNN` can count and detect important substructures, Table 2 that it can transfer to out-of-distribution and larger datasets, and Tables 4, 5 that it is effective in downstream tasks. Overall, the results provide concrete evidence of the generic nature of `Moment-GNN`, which overcomes the need for fine-tuning based on the most pivotal subgraph type for each specific task. Further experiments on the full ZINC dataset, along with additional baseline comparisons and discussions can be found in Appendix M.

## 8 CONCLUSION

In this paper, we analyzed the representation power of local, message-passing GNNs with respect to their ability to generate representations that count certain important graph substructures. To do so we studied GNNs with random node inputs and showed that proper choices of activation and normalization functions enable the generation of powerful equivariant representations in the output. These representations are provably capable of counting cycles with 3 to 8 nodes, cliques with 3 to 4 nodes, several quasi-cliques, and connected components within a graph. They can also generalize to arbitrary out-of-distribution graphs. Our analysis is constructive and designed a generic architecture that demonstrated remarkable performance in the tasks of subgraph detection, subgraph counting, graph classification and logP prediction.

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

## A  RELATED WORK

The expressive power of Graph Neural Networks (GNNs) was initially explored by Xu et al. (2019); Morris et al. (2019). These studied the expressive power of GNNs in terms of performing the graph isomorphism test, i.e., distinguishing different graphs. They identified a connection between the Weisfeiler-Lehman (WL) test Weisfeiler & Leman (1968) and the message-passing operations in GNNs. Using this connection they proved that GNNs with degree-related inputs are, at most, as powerful as the WL test. Arvind et al. (2020); Chen et al. (2020) extended these results to the task of substructure counting, and showed that GNNs with degree-related inputs cannot count important substructures such as triangles.

Expanding on these findings, several works Maron et al. (2018); Murphy et al. (2019); Azizian et al. (2020); Morris et al. (2020); Geerts & Reutter (2021); Giusti et al. (2022) introduced high-order GNN models that utilize tensor representations and capture k-tuple and k-subgraph information. These high-order GNNs offer improved counting capabilities but come with increased computational and memory complexity. Maron et al. (2019); Chen et al. (2019) reduce the computational complexity of high-order GNNs by working directly with matrix or tensor multiplications while maintaining high levels of representation power.

Another research direction, addressing the limitations identified by Xu et al. (2019); Morris et al. (2019), focuses on accurately counting substructures by providing GNNs with unique node identifiers Loukas (2019); Abboud et al. (2021); Sato et al. (2021). Notably, Abboud et al. (2021); Sato et al. (2021) employ random features as input to GNNs, resulting in enhanced function approximation. However, this approach sacrifices the essential property of permutation equivariance, which is fundamental in graph learning. Alternatively, Vignac et al. (2020) incorporate global graph information at the node level, leading to improved substructure counting capabilities but with increased computational and memory requirements.

Tahmasebi et al. (2020); You et al. (2021); Bouritsas et al. (2022); Barceló et al. (2021); Bevilacqua et al. (2021); Zhang & Li (2021); Zhao et al. (2021); Bodnar et al. (2021) directly augment GNNs with structural information to enhance their representation power. Some of them, as Tahmasebi et al. (2020); You et al. (2021); Bouritsas et al. (2022) directly augment GNNs with structural or spectral Lim et al. (2022) information, that learn from non-message-passing mechanisms, to enhance their representation power and empirical performance. Others as Zhang & Li (2021); Zhao et al. (2021); Bodnar et al. (2021) first identify important subgraphs and then they design message-passing mechanisms that operate according to these subgraphs. This way, they learn local subgraph representations that are used to extract global graph embeddings. Transformer GNN architectures have been proposed Veličković et al. (2018); Rampášek et al. (2022) with increased expressive power. Finally, the statistical guarantees GNNs in link prediction are studied in Chung et al. (2024).

## B  MESSAGE-PASSING GRAPH NEURAL NETWORKS

The most typical Graph Neural Network (GNN) architecture is the message-passing GNN (MPGNN) (Kipf & Welling, 2016; Gilmer et al., 2017; Xu et al., 2019), which is usually defined as follows:

$$\boldsymbol{x}_v^{(l)} = g^{(l-1)} \left( \boldsymbol{x}_v^{(l-1)}, f^{(l-1)} \left( \left\{ \boldsymbol{x}_u^{(l-1)} : u \in \mathcal{N}(v) \right\} \right) \right), \tag{17}$$

where $\mathcal{N}(v)$ is the neighborhood of vertex $v$, i.e., $u \in \mathcal{N}(v)$ if and only if $(u, v) \in \mathcal{E}$. The function $f^{(l)}$ aggregates information from the multiset of neighboring signals, whereas $g^{(l)}$ combines the signal of each vertex with the aggregated one from the neighboring vertices.

### B.1 From message-passing to graph convolution

When $f$ is the summation function and $g$ is concatenation followed by multivariate linear function, 17 becomes:

$$\boldsymbol{x}_v^{(l)} = \boldsymbol{A}^{(l-1)}\boldsymbol{x}_v^{(l-1)} + \boldsymbol{B}^{(l-1)} \sum_{u \in \mathcal{N}(v)} \boldsymbol{x}_u^{(l-1)}, \tag{18}$$

which we can write in matrix form as:

$$\boldsymbol{X}^{(l)} = \boldsymbol{X}^{(l-1)}\boldsymbol{A}^{(l-1)^T} + \boldsymbol{S}\boldsymbol{X}^{(l-1)}\boldsymbol{B}^{(l-1)^T}, \tag{19}$$

where $\boldsymbol{S} \in \{0,1\}^{N \times N}$ is the graph adjacency and the $v-$th row of $\boldsymbol{X}^{(l-1)}$ is $\boldsymbol{X}^{(l-1)}[v,:] = \boldsymbol{x}_v^{(l-1)^T}$. For 2 MPNN the recursive formula gives:

$$\boldsymbol{X}^{(1)} = \boldsymbol{X}\boldsymbol{A}^{(0)^T} + \boldsymbol{S}\boldsymbol{X}\boldsymbol{B}^{(0)^T} \tag{20}$$

$$\boldsymbol{X}^{(2)} = \boldsymbol{X}^{(1)}\boldsymbol{A}^{(1)^T} + \boldsymbol{S}\boldsymbol{X}^{(1)}\boldsymbol{B}^{(1)^T} \tag{21}$$

$$= \boldsymbol{X}\boldsymbol{A}^{(0)^T}\boldsymbol{A}^{(1)^T} + \boldsymbol{S}\boldsymbol{X}\left(\boldsymbol{B}^{(0)^T}\boldsymbol{A}^{(1)^T} + \boldsymbol{A}^{(0)^T}\boldsymbol{B}^{(1)^T}\right) + \boldsymbol{S}^2\boldsymbol{X}\boldsymbol{B}^{(0)^T}\boldsymbol{B}^{(1)^T} \tag{22}$$

$$= \boldsymbol{X}\boldsymbol{H}_0 + \boldsymbol{S}\boldsymbol{X}\boldsymbol{H}_1 + \boldsymbol{S}^2\boldsymbol{X}\boldsymbol{H}_2, \tag{23}$$

where $\boldsymbol{X} = \boldsymbol{X}^{(0)}$. Following the previous analysis $K$ MPNN layers yield:

$$\boldsymbol{Z} = \boldsymbol{X}^{(K)} = \sum_{k=0}^{K} \boldsymbol{S}\boldsymbol{X}\boldsymbol{H}_k. \tag{24}$$

After the $K-$th layer we can also add a pointwise activation function $\sigma\left(\cdot\right)$ and a normalization layer $\rho\left[\cdot\right]$ which are the norm in deep learning. Then

$$\boldsymbol{Y} = \rho\left[\sigma\left(\boldsymbol{Z}\right)\right] = \rho\left[\sigma\left(\sum_{k=0}^{K} \boldsymbol{S}\boldsymbol{X}\boldsymbol{H}_k\right)\right]. \tag{25}$$

Overall, the above expression is an MPNN defined in 17, where $f^{(l)}$ is the summation function, and $g^{(l)}$ is the linear function for $l = 1, \ldots K-1$ and the MLP followed by a normalization layer for $l = K$.

### B.2 The local GNN architecture

In this paper, we study a cascade of MPNN layers defined in 25, i.e.,

$$\boldsymbol{X}^{(l+1)} = \rho\left[\sigma\left(\boldsymbol{Z}\right)\right] = \rho\left[\sigma\left(\sum_{k=0}^{K} \boldsymbol{S}\boldsymbol{X}^{(l)}\boldsymbol{H}_k\right)\right]. \tag{26}$$

The type of pointwise nonlinearities that we study are the Rectified Linear Unit (ReLU), and elementwise powers $\sigma\left(\cdot\right) = \left(\cdot\right)^m$. We also consider the following types of normalization layers:

$$\rho\left[\cdot\right] = \mathbb{E}\left[\left(\cdot\right) \odot \left(\boldsymbol{1}^T\boldsymbol{z}\right)^m\right] \tag{27}$$

$$\rho\left[\cdot\right] = \mathbb{E}\left[\left(\cdot\right) \odot \left(\boldsymbol{1}^T\boldsymbol{z}^m\right)\right] \tag{28}$$

$$\rho\left[\cdot\right] = \left(\cdot\right) \odot \mathbb{E}\left[\boldsymbol{z}^m\right] \tag{29}$$

where $\mathbb{E}$ is the expectation operator, $\odot$ is the Hadamard (elementwise) product, and all the powers are also elementwise.

## C Tensor Algebra Preliminaries

To facilitate our analysis, we briefly present some tensor algebra preliminaries and refer the reader to Sidiropoulos et al. (2017); Kolda & Bader (2009) for further details.

A $N$-order tensor $\underline{\boldsymbol{X}} \in \mathbb{R}^{I_1 \times I_2 \times \cdots \times I_N}$ is an $N$-way array indexed by $i_1, i_2, \ldots, i_N$ with elements $\underline{\boldsymbol{X}}(i_1, i_2, \ldots, i_N)$. It consists of $N$ types of modes: $\underline{\boldsymbol{X}}(:, i_2, \ldots, i_N)$, $\underline{\boldsymbol{X}}(i_1, :, \ldots, i_N), \ldots, \underline{\boldsymbol{X}}(i_1, i_2, \ldots, :)$.

A rank-one tensor $\underline{\boldsymbol{Z}} \in \mathbb{R}^{I_1 \times I_2 \times \cdots \times I_N}$ is the outer product of $N$ vectors defined as:

$$\underline{\boldsymbol{Z}} = \boldsymbol{a}_1 \circ \boldsymbol{a}_2 \circ \cdots \circ \boldsymbol{a}_N, \tag{30}$$

where $\boldsymbol{a}_1 \in \mathbb{R}^{I_1}$, $\boldsymbol{a}_2 \in \mathbb{R}^{I_2}, \ldots, \boldsymbol{a}_N \in \mathbb{R}^{I_N}$ and $\circ$ denotes the outer product. The elementwise formula of the above expression is:

$$\underline{\boldsymbol{Z}}(i_1, i_2, \ldots, i_N) = \boldsymbol{a}_1(i_1)\boldsymbol{a}_2(i_2) \cdots \boldsymbol{a}_N(i_N), \ \forall i_1, i_2, \ldots, i_N, \tag{31}$$

Any tensor can be realized as a sum of $N$-way outer products (rank one tensors), i.e.

$$\underline{\boldsymbol{X}} = \sum_{f=1}^{F} \boldsymbol{a}_1^f \circ \boldsymbol{a}_2^f \circ \cdots \circ \boldsymbol{a}_N^f. \tag{32}$$

The above expression represents the *canonical polyadic decomposition* (CPD) or *parallel factor analysis* (PARAFAC) Harshman & Lundy (1994) of a tensor. The CPD elementwise representation is:

$$\underline{\boldsymbol{X}}(i, j, k) = \sum_{f=1}^{F} \boldsymbol{A}_1(i_1, f)\boldsymbol{A}_2(i_2, f) \cdots \boldsymbol{A}_N(i_N, f), \tag{33}$$

where $\boldsymbol{A_n} = [\boldsymbol{a}_n^1, \boldsymbol{a}_n^2, \ldots, \boldsymbol{a}_n^F] \in \mathbb{R}^{I_n \times F}$, $n = 1, \ldots, N$ are called the low rank factors of the tensor. A tensor can be fully characterized by its latent factors, so we can represent a tensor by its CPD model as:

$$\underline{\boldsymbol{X}} = [\![\boldsymbol{A}_1, \boldsymbol{A}_2, \ldots, \boldsymbol{A}_N]\!]. \tag{34}$$

A tensor can be also represented as a set of matrices, by fixing all the modes but two as:

$$\underline{\boldsymbol{X}}[:, :, i_3, \ldots, i_N] = \boldsymbol{A}_1 \left( \mathrm{Diag}\left(\boldsymbol{A}_3(i_3, :)\right) \odot \cdots \odot \mathrm{Diag}\left(\boldsymbol{A}_N(i_N, :)\right) \right) \boldsymbol{A}_2^T, \tag{35}$$

where $\mathrm{Diag}\left(\boldsymbol{A}_n(i_n, :)\right)$ is the diagonal matrix with diagonal equal to $\boldsymbol{A}_N(i_n, :)$.

An important operation in tensor analytics is the *mode product* which multiplies a matrix or a vector to a tensor in a single mode. A joint mode-1, mode-2,..., mode-N product of a tensor is represented as follows:

$$\tilde{\underline{\boldsymbol{X}}} = \underline{\boldsymbol{X}} \times_1 \boldsymbol{P}_1 \times_2 \boldsymbol{P}_2 \cdots \times_N \boldsymbol{P}_N \tag{36}$$

where "$\times_1$" denotes the operation that multiplies each column of $\underline{\boldsymbol{X}}$ with $\boldsymbol{P}_1$, "$\times_2$" denotes multiplying each row of $\underline{\boldsymbol{X}}$ with $\boldsymbol{P}_2$, and "$\times_N$" denotes multiplying each $N-$mode of $\underline{\boldsymbol{X}}$ with $\boldsymbol{P}_N$. The mode product is reflected in the CPD model of the tensor, i.e., the outcome of 36 results in a tensor $\tilde{\underline{\boldsymbol{X}}}$ with CPD:

$$\tilde{\underline{\boldsymbol{X}}} = [\![\boldsymbol{P}_1 \boldsymbol{A}_1, \boldsymbol{P}_2 \boldsymbol{A}_2, \ldots, \boldsymbol{P}_N \boldsymbol{A}_N]\!],$$

## D  STUDYING GNNS WITH RANDOM INPUT

To analyze the representation power of GNNs we study them with a random input $\boldsymbol{x} = [x_1, x_2, \ldots, x_N]^T \ in \mathbb{R}^N$ that is characterized by the following moments:

$$\mathbb{E}[\boldsymbol{x}] = \boldsymbol{0}, \quad \mathbb{E}[\boldsymbol{x}\boldsymbol{x}^T] = \boldsymbol{I}, \quad \mathbb{E}[\boldsymbol{x} \circ \boldsymbol{x} \circ \boldsymbol{x}] = \underline{\boldsymbol{I}}, \quad \mathbb{E}[\boldsymbol{x} \circ \boldsymbol{x} \circ \cdots \circ \boldsymbol{x}] = \underline{\boldsymbol{I}}. \tag{37}$$

The characteristic function of $x_i$ takes the form:

$$\phi_{x_i}(t) = \mathbb{E}\left[e^{jx_i t}\right] = \mathbb{E}\left[\sum_{l=0}^{\infty} \frac{(jx_i t)^l}{l!}\right] = \sum_{l=0}^{\infty} \frac{j^l \, \mathbb{E}\left[(x_i t)^l\right]}{l!} = 1 + \sum_{l=2}^{\infty} \frac{j^l \, (t^l)}{l!} \tag{38}$$

$$= \sum_{l=0}^{\infty} \frac{(j\,t)^l}{l!} - jt = \left(e^{jt} - jt\right) = e^{j\,t} - jt \tag{39}$$

As a result the joint characteristic function of $\boldsymbol{x}$ is $\phi_{\boldsymbol{x}}(t_1, \ldots, t_N) = \prod_{i=1}^{N} \left(e^{jt_i} - jt_i\right)$.

As mentioned in the main body of the paper, the first layer of a GNN with random input consists of an $F$ set of graph filters that are followed by point-wise activation functions and normalization layers.

$$\boldsymbol{z} = \sum_{k=0}^{K} h_k \boldsymbol{S}^k \boldsymbol{x} = \boldsymbol{H}(\boldsymbol{S})\boldsymbol{x}, \quad \boldsymbol{y} = \rho\left[\sigma(\boldsymbol{z})\right], \tag{40}$$

### D.1 Second order moment

Since $\boldsymbol{x}$ is a random vector, so is $\boldsymbol{z}$ with covariance matrix:

$$\mathbb{E}\left[\boldsymbol{z}\boldsymbol{z}^T\right] = \mathbb{E}\left[\sum_{k=0}^{K} h_k \boldsymbol{S}^k \boldsymbol{x}\boldsymbol{x}^T \sum_{m=0}^{K} h_m \boldsymbol{S}^{m^T}\right] = \sum_{k=0}^{K} h_k \boldsymbol{S}^k \mathbb{E}\left[\boldsymbol{x}\boldsymbol{x}^T\right] \sum_{m=0}^{K} h_m \boldsymbol{S}^m \qquad (41)$$

$$= \sum_{k=0}^{K} h_k \boldsymbol{S}^k \sum_{m=0}^{K} h_m \boldsymbol{S}^m = \boldsymbol{H}\left(\boldsymbol{S}\right)\boldsymbol{H}\left(\boldsymbol{S}\right) = \sum_{k=0}^{K}\sum_{m=0}^{K} h_k h_m \boldsymbol{S}^k \boldsymbol{S}^m \qquad (42)$$

$$= \sum_{k,m=0}^{K} h_{k,m} \boldsymbol{S}^{k+m} = \sum_{k=0}^{2K} \tilde{h}_k \boldsymbol{S}^k = \tilde{\boldsymbol{H}}\left(\boldsymbol{S}\right). \qquad (43)$$

So the covariance of $\boldsymbol{z}$ is a graph filter with $2K+1$ parameters that have $K+1$ degrees of freedom. The parameters of this graph filter are derived from $\tilde{\boldsymbol{h}} = \boldsymbol{h} * \boldsymbol{h}$, where $*$ is the convolution operation.

From the covariance of $z$ we can linearly derive two permutation equivariant features for each vertex. In fact, if we sum up all the rows of $\mathbb{E}\left[\boldsymbol{z}\boldsymbol{z}^T\right]$ we get:

$$\mathbb{E}\left[\boldsymbol{z}\boldsymbol{z}^T\right]\boldsymbol{1} = \tilde{\boldsymbol{H}}\left(\boldsymbol{S}\right)\boldsymbol{1} = \sum_{k,m=0}^{K} h_{k,m}\boldsymbol{S}^{k+m}\boldsymbol{1} = \sum_{k=0}^{2K}\tilde{h}_k \boldsymbol{S}^k \boldsymbol{1}, \qquad (44)$$

which is a linear combination of the $k-$hop degrees of $\boldsymbol{S}$. To derive this feature according to 40 we choose $\sigma$ to be the identity function and $\rho\left(\cdot\right) = \mathbb{E}\left[(\cdot)\odot\left(\boldsymbol{1}^T\boldsymbol{z}\right)\right]$. For these choices, we get:

$$\boldsymbol{y} = \rho\left[\sigma\left(\boldsymbol{z}\right)\right] = \mathbb{E}\left[\boldsymbol{z}\left(\boldsymbol{1}^T\boldsymbol{z}\right)\right] = \mathbb{E}\left[\boldsymbol{z}\boldsymbol{z}^T\right]\boldsymbol{1} = \tilde{\boldsymbol{H}}\left(\boldsymbol{S}\right)\boldsymbol{1} = \sum_{k,m=0}^{K} h_{k,m}\boldsymbol{S}^{k+m}\boldsymbol{1}. \qquad (45)$$

The second equivariant feature is the diagonal of the covariance matrix, i.e., the variance of $\boldsymbol{z}$. If we choose $\sigma$ to be the elementwise square function and $\rho$ to be the expectation operator we compute:

$$\boldsymbol{y} = \rho\left[\sigma\left(\boldsymbol{z}\right)\right] = \mathbb{E}\left[\boldsymbol{z}^2\right] = \operatorname{diag}\left(\mathbb{E}\left[\boldsymbol{z}\boldsymbol{z}^T\right]\right) = \operatorname{diag}\left(\boldsymbol{H}\left(\boldsymbol{S}\right)\boldsymbol{H}\left(\boldsymbol{S}\right)\right) = \left(\boldsymbol{H}\left(\boldsymbol{S}\right)\odot\boldsymbol{H}\left(\boldsymbol{S}\right)\right)\boldsymbol{1}, \quad (46)$$

where $\operatorname{diag}(\boldsymbol{A}) \in \mathbb{R}^N$ is the vector with the diagonal of $\boldsymbol{A} \in \mathbb{R}^{N\times N}$, $\odot$ denotes the Hadamard (elementwise) product operation, and the last equality comes from the following property:

$$\operatorname{diag}\left(\boldsymbol{A}\boldsymbol{B}\right) = \left(\boldsymbol{A}\odot\boldsymbol{B}\right)\boldsymbol{1}, \qquad (47)$$

for square matrices $\boldsymbol{A}$, $\boldsymbol{B}$. Expanding 46 yields the following expression:

$$\boldsymbol{y} = \sum_{k,m} h_{k,m}\operatorname{diag}\left(\boldsymbol{S}^{k+m}\right) = \sum_{k,m} h_{k,m}\left(\boldsymbol{S}^k \odot \boldsymbol{S}^m\right)\boldsymbol{1}, \qquad (48)$$

that computes the number of $k+m$ closed paths in the graph with adjacency matrix $\boldsymbol{S}$.

### D.2 Third-order moment

If we choose $\sigma$ to be the elementwise cubic function and $\rho$ to be the expectation operator we can compute deterministic equivariant node features that measure the skewness of $\boldsymbol{z}$.

$$\boldsymbol{y}_1 = \rho\left[\sigma\left(\boldsymbol{z}\right)\right] = \mathbb{E}\left[\boldsymbol{z}^3\right] = \operatorname{superdiag}\left(\mathbb{E}\left[\boldsymbol{z}\circ\boldsymbol{z}\circ\boldsymbol{z}\right]\right) = \left(\boldsymbol{H}\left(\boldsymbol{S}\right)\odot\boldsymbol{H}\left(\boldsymbol{S}\right)\odot\boldsymbol{H}\left(\boldsymbol{S}\right)\right)\boldsymbol{1}. \quad (49)$$

where $\operatorname{superdiag}\left(\cdot\right) \in \mathbb{R}^N$ is the superdiagonal of the tensor. In particular,

$$\mathbb{E}\left[\boldsymbol{z}\circ\boldsymbol{z}\circ\boldsymbol{z}\right]\left(i,j,k\right) = \sum_{n=1}^{N} \boldsymbol{H}\left(\boldsymbol{S}\right)\left(i,n\right)\boldsymbol{H}\left(\boldsymbol{S}\right)\left(j,n\right)\boldsymbol{H}\left(\boldsymbol{S}\right)\left(k,n\right) \qquad (50)$$

and

$$\text{superdiag}\left(\mathbb{E}\left[\boldsymbol{z}\circ\boldsymbol{z}\circ\boldsymbol{z}\right]\right)(i)=\mathbb{E}\left[\boldsymbol{z}\circ\boldsymbol{z}\circ\boldsymbol{z}\right](i,i,i)$$

$$=\sum_{n=1}^{N}\boldsymbol{H}\left(\boldsymbol{S}\right)(i,n)\boldsymbol{H}\left(\boldsymbol{S}\right)(i,n)\boldsymbol{H}\left(\boldsymbol{S}\right)(i,n)=\left(\boldsymbol{H}\left(\boldsymbol{S}\right)(i,:)\odot\boldsymbol{H}\left(\boldsymbol{S}\right)(i,:)\odot\boldsymbol{H}\left(\boldsymbol{S}\right)(i,:)\right)\mathbf{1}$$

$$(51)$$

As a result, 9 can be cast as follows:

$$\boldsymbol{y}_1=\left(\sum_{k=0}^{K}h_k\boldsymbol{S}^k\odot\sum_{l=0}^{K}h_l\boldsymbol{S}^l\odot\sum_{m=0}^{K}h_m\boldsymbol{S}^m\right)\mathbf{1}=\sum_{k,l,m=0}^{K}h_{k,l,m}\left(\boldsymbol{S}^k\odot\boldsymbol{S}^l\odot\boldsymbol{S}^m\right)\mathbf{1}. \quad (52)$$

The expression in 11 can be derived from 4 when $\sigma$ is the elementwise square function, and $\rho\left(\cdot\right)=\mathbb{E}\left[\left(\cdot\right)\odot\left(\mathbf{1}^T\boldsymbol{z}\right)\right]$.

$$\mathbb{E}\left[\boldsymbol{z}\circ\boldsymbol{z}\circ\boldsymbol{z}\right]\times_3\mathbf{1}^T=\left[\!\left[\boldsymbol{H}\left(\boldsymbol{S}\right),\boldsymbol{H}\left(\boldsymbol{S}\right),\mathbf{1}^T\boldsymbol{H}\left(\boldsymbol{S}\right)\right]\!\right]=\boldsymbol{H}\left(\boldsymbol{S}\right)\text{Diag}\left(\mathbf{1}^T\boldsymbol{H}\left(\boldsymbol{S}\right)\right)\boldsymbol{H}\left(\boldsymbol{S}\right) \quad (53)$$

$$\text{diag}\left(\mathbb{E}\left[\boldsymbol{z}\circ\boldsymbol{z}\circ\boldsymbol{z}\right]\times_3\mathbf{1}^T\right)=\text{diag}\left(\boldsymbol{H}\left(\boldsymbol{S}\right)\text{Diag}\left(\mathbf{1}^T\boldsymbol{H}\left(\boldsymbol{S}\right)\right)\boldsymbol{H}\left(\boldsymbol{S}\right)\right)=\left(\boldsymbol{H}\left(\boldsymbol{S}\right)\odot\boldsymbol{H}\left(\boldsymbol{S}\right)\right)\boldsymbol{H}\left(\boldsymbol{S}\right)\mathbf{1},$$

$$(54)$$

where $\text{diag}\left(\boldsymbol{A}\right)\in\mathbb{R}^N$ is the vector that contains the diagonal of $\boldsymbol{A}\in\mathbb{R}^{N\times N}$ and is different than $\text{Diag}(\cdot)$. The last equality comes from the following property that is being used a lot in this paper:

$$\text{diag}\left(\boldsymbol{A}\text{Diag}\left(\boldsymbol{C}\right)\boldsymbol{B}^T\right)=\left(\boldsymbol{A}\odot\boldsymbol{B}\right)\text{diag}\left(\boldsymbol{C}\right), \quad (55)$$

Overall, 11 takes the form:

$$\boldsymbol{y}_2=\left(\sum_{k=0}^{K}h_k\boldsymbol{S}^k\odot\sum_{l=0}^{K}h_l\boldsymbol{S}^l\right)\sum_{m=0}^{K}h_m\boldsymbol{S}^m\mathbf{1}=\sum_{k,l,m=0}^{K}h_{k,l,m}\left(\boldsymbol{S}^k\odot\boldsymbol{S}^l\right)\boldsymbol{S}^m\mathbf{1}. \quad (56)$$

### D.3 FOURTH-ORDER MOMENT

The fourth-order moment is defined as follows:

$$\mathbb{E}\left[\boldsymbol{z}\circ\boldsymbol{z}\circ\boldsymbol{z}\circ\boldsymbol{z}\right]=\left[\!\left[\boldsymbol{H}\left(\boldsymbol{S}\right),\boldsymbol{H}\left(\boldsymbol{S}\right),\boldsymbol{H}\left(\boldsymbol{S}\right),\boldsymbol{H}\left(\boldsymbol{S}\right)\right]\!\right]=\sum_{k,l,m,n=0}^{K}h_{k,l,m,n}\left[\!\left[\boldsymbol{S}^k,\boldsymbol{S}^l,\boldsymbol{S}^m,\boldsymbol{S}^n\right]\!\right] \quad (57)$$

We can compute the superdiagonal as:

$$\boldsymbol{y}=\mathbb{E}\left[\boldsymbol{z}^4\right]=\text{superdiag}\left(\mathbb{E}\left[\boldsymbol{z}\circ\boldsymbol{z}\circ\boldsymbol{z}\circ\boldsymbol{z}\right]\right)=\left(\boldsymbol{H}\left(\boldsymbol{S}\right)\odot\boldsymbol{H}\left(\boldsymbol{S}\right)\odot\boldsymbol{H}\left(\boldsymbol{S}\right)\odot\boldsymbol{H}\left(\boldsymbol{S}\right)\right)\mathbf{1} \quad (58)$$

However, $\mathbb{E}\left[\boldsymbol{z}^4\right]$ is not the only equivariant feature we can derive from the fourth moment tensor $\mathbb{E}\left[\boldsymbol{z}\circ\boldsymbol{z}\circ\boldsymbol{z}\circ\boldsymbol{z}\right]$. In particular, we can compute:

$$\begin{aligned}
\mathbb{E}\left[\boldsymbol{z}^2\left(\mathbf{1}^T\boldsymbol{z}\right)^2\right]&=\text{diag}\left(\mathbb{E}\left[\boldsymbol{z}\circ\boldsymbol{z}\circ\boldsymbol{z}\circ\boldsymbol{z}\right]\times_3\mathbf{1}^T\times_4\mathbf{1}^T\right)\\
&=\text{diag}\left(\left[\!\left[\boldsymbol{H}\left(\boldsymbol{S}\right),\boldsymbol{H}\left(\boldsymbol{S}\right),\boldsymbol{H}\left(\boldsymbol{S}\right),\boldsymbol{H}\left(\boldsymbol{S}\right)\right]\!\right]\times_3\mathbf{1}\times_4\mathbf{1}\right)\\
&=\text{diag}\left(\boldsymbol{H}\left(\boldsymbol{S}\right)\left(\text{Diag}\left(\mathbf{1}^T\boldsymbol{H}\left(\boldsymbol{S}\right)\right)\text{Diag}\left(\mathbf{1}^T\boldsymbol{H}\left(\boldsymbol{S}\right)\right)\right)\boldsymbol{H}\left(\boldsymbol{S}\right)\right)\\
&=\text{diag}\left(\left[\!\left[\boldsymbol{H}\left(\boldsymbol{S}\right),\boldsymbol{H}\left(\boldsymbol{S}\right),\mathbf{1}^T\boldsymbol{H}\left(\boldsymbol{S}\right),\mathbf{1}^T\boldsymbol{H}\left(\boldsymbol{S}\right)\right]\!\right]\right)\\
&=\text{diag}\left(\boldsymbol{H}\left(\boldsymbol{S}\right)\left(\text{Diag}\left(\mathbf{1}^T\boldsymbol{H}\left(\boldsymbol{S}\right)\right)\text{Diag}\left(\mathbf{1}^T\boldsymbol{H}\left(\boldsymbol{S}\right)\right)\right)\boldsymbol{H}\left(\boldsymbol{S}\right)\right)\\
&=\left(\boldsymbol{H}\left(\boldsymbol{S}\right)\odot\boldsymbol{H}\left(\boldsymbol{S}\right)\right)\left(\boldsymbol{H}\left(\boldsymbol{S}\right)\mathbf{1}\odot\boldsymbol{H}\left(\boldsymbol{S}\right)\mathbf{1}\right)\\
&=\sum_{k,l,m,n=0}^{K}h_{k,l,m,n}\left(\boldsymbol{S}^k\odot\boldsymbol{S}^l\right)\left(\boldsymbol{S}^m\mathbf{1}\odot\boldsymbol{S}^n\mathbf{1}\right), \quad (59)
\end{aligned}$$

which can be implemented when $\sigma$ is the elementwise square function and $\rho\left(\cdot\right) = \mathbb{E}\left[\left(\cdot\right) \odot \left(\mathbf{1}^T \boldsymbol{z}\right)^2\right]$. We can also compute:

$$
\begin{aligned}
\mathbb{E}\left[\boldsymbol{z}^3\left(\mathbf{1}^T\boldsymbol{z}\right)\right] &= \operatorname{superdiag}\left(\mathbb{E}\left[\boldsymbol{z} \circ \boldsymbol{z} \circ \boldsymbol{z} \circ \boldsymbol{z}\right] \times_4 \mathbf{1}^T\right) \\
&= \operatorname{superdiag}\left(\llbracket \boldsymbol{H}\left(\boldsymbol{S}\right), \boldsymbol{H}\left(\boldsymbol{S}\right), \boldsymbol{H}\left(\boldsymbol{S}\right), \boldsymbol{H}\left(\boldsymbol{S}\right)\rrbracket \times_4 \mathbf{1}^T\right) \\
&= \operatorname{superdiag}\left(\llbracket \boldsymbol{H}\left(\boldsymbol{S}\right), \boldsymbol{H}\left(\boldsymbol{S}\right), \boldsymbol{H}\left(\boldsymbol{S}\right), \mathbf{1}^T\boldsymbol{H}\left(\boldsymbol{S}\right)\rrbracket\right) \\
&= \left(\boldsymbol{H}\left(\boldsymbol{S}\right) \odot \boldsymbol{H}\left(\boldsymbol{S}\right) \odot \boldsymbol{H}\left(\boldsymbol{S}\right)\right)\boldsymbol{H}\left(\boldsymbol{S}\right)\mathbf{1} \\
&= \left(\boldsymbol{H}\left(\boldsymbol{S}\right) \odot \boldsymbol{H}\left(\boldsymbol{S}\right)\right)\left(\boldsymbol{H}\left(\boldsymbol{S}\right)\mathbf{1} \odot \boldsymbol{H}\left(\boldsymbol{S}\right)\mathbf{1}\right) \\
&= \sum_{k,l,m,n=0}^K h_{k,l,m,n}\left(\boldsymbol{S}^k \odot \boldsymbol{S}^l \odot \boldsymbol{S}^m\right)\boldsymbol{S}^n\mathbf{1},
\end{aligned}
\tag{60}
$$

which can be implemented when $\sigma$ is the elementwise cubic function and $\rho\left(\cdot\right) = \mathbb{E}\left[\left(\cdot\right) \odot \left(\mathbf{1}^T\boldsymbol{z}\right)\right]$. Finally, we can compute $\mathbb{E}\left[\boldsymbol{z}^2\left(\mathbf{1}^T\boldsymbol{z}^2\right)\right] = \mathbb{E}\left[\boldsymbol{z}^2 \circ \boldsymbol{z}^2\right]\mathbf{1}$. To derive the exact expression we analyze the following

$$
\begin{aligned}
\mathbb{E}\left[\boldsymbol{z}^2 \circ \boldsymbol{z}^2\right]\left[i,j\right] &= \mathbb{E}\left[\boldsymbol{z} \circ \boldsymbol{z} \circ \boldsymbol{z} \circ \boldsymbol{z}\right]\left[i,i,j,j\right] \\
&= \sum_{k,l,m,n=0}^K h_{k,l,m,n}\llbracket \boldsymbol{S}^k, \boldsymbol{S}^l, \boldsymbol{S}^m, \boldsymbol{S}^n\rrbracket\left[i,i,j,j\right] \\
&= \sum_{k,l,m,n=0}^K h_{k,l,m,n}\sum_{n=1}^N \boldsymbol{S}^k\left[i,n\right]\boldsymbol{S}^l\left[i,n\right]\boldsymbol{S}^m\left[j,n\right]\boldsymbol{S}^n\left[j,n\right] \\
&= \sum_{k,l,m,n=0}^K h_{k,l,m,n}\sum_{n=1}^N \left(\boldsymbol{S}^k \odot \boldsymbol{S}^l\right)\left[i,n\right]\left(\boldsymbol{S}^m \odot \boldsymbol{S}^n\right)\left[j,n\right]
\end{aligned}
\tag{61}
$$

As a result:

$$
\mathbb{E}\left[\boldsymbol{z}^2 \circ \boldsymbol{z}^2\right] = \sum_{k,l,m,n=0}^K h_{k,l,m,n}\left(\boldsymbol{S}^k \odot \boldsymbol{S}^l\right)\left(\boldsymbol{S}^m \odot \boldsymbol{S}^n\right)
\tag{62}
$$

and we compute:

$$
\mathbb{E}\left[\boldsymbol{z}^2\left(\mathbf{1}^T\boldsymbol{z}^2\right)\right] = \mathbb{E}\left[\boldsymbol{z}^2 \circ \boldsymbol{z}^2\right]\mathbf{1} = \sum_{k,l,m,n=0}^K h_{k,l,m,n}\left(\boldsymbol{S}^k \odot \boldsymbol{S}^l\right)\left(\boldsymbol{S}^m \odot \boldsymbol{S}^n\right)\mathbf{1},
\tag{63}
$$

which can be implemented when $\sigma$ is the elementwise square function and $\rho\left(\cdot\right) = \mathbb{E}\left[\left(\cdot\right) \odot \left(\mathbf{1}^T\boldsymbol{z}^2\right)\right]$.

### D.4 HIGHER-ORDER MOMENTS

When $\sigma$ is the $m$-th moment of $z$ is an $m$-th-order tensor and takes the form:

$$
\mathbb{E}\left[\boldsymbol{z} \circ \cdots \circ \boldsymbol{z}\right] = \llbracket \boldsymbol{H}\left(\boldsymbol{S}\right), \ldots, \boldsymbol{H}\left(\boldsymbol{S}\right)\rrbracket = \sum_{i_1,\ldots,i_m=0}^K h_{i_1,\ldots,i_m}\llbracket \boldsymbol{S}^{i_1}, \ldots, \boldsymbol{S}^{i_m}\rrbracket,
\tag{64}
$$

from which we can extract the equivariant individual moment:

$$
\mathbb{E}\left[\boldsymbol{z}^m\right] = \left(\boldsymbol{H}\left(\boldsymbol{S}\right) \odot \cdots \odot \boldsymbol{H}\left(\boldsymbol{S}\right)\right)\mathbf{1} = \sum_{i_1,\ldots,i_m=0}^K h_{i_1,\ldots,i_m}\left(\boldsymbol{S}^{i_1} \odot \cdots \odot \boldsymbol{S}^{i_m}\right)\mathbf{1},
\tag{65}
$$

which can be derived by 2 when $\sigma$ is the elementwise $m$-th power and $\rho$ is the expectation operator. We can also extract other equivariant node features by considering multiple GNN layers of 2 type, summing up certain modes of the tensor and applying diagonal or super-diagonal operators.

Overall a single layer GNN (which corresponds to $K$ MPNN layers, as defined in 26, can compute the following outputs:

$$\boldsymbol{y} = \sum_{i_1,\dots,i_m=0}^{K} h_{i_1,\dots,i_m} \left( \boldsymbol{S}^{i_1} \odot \cdots \odot \boldsymbol{S}^{i_k} \right) \left( \boldsymbol{S}^{i_{k+1}} \odot \cdots \odot \boldsymbol{S}^{i_m} \right) \mathbf{1} \tag{66}$$

$$\boldsymbol{y} = \sum_{i_1,\dots,i_m=0}^{K} h_{i_1,\dots,i_m} \left( \boldsymbol{S}^{i_1} \odot \cdots \odot \boldsymbol{S}^{i_k} \right) \left( \boldsymbol{S}^{i_{k+1}} \mathbf{1} \odot \cdots \odot \boldsymbol{S}^{i_m} \mathbf{1} \right), \tag{67}$$

which are derived by the $m-$th moment of $\boldsymbol{z}$. The expression in using $\sigma\left(\cdot\right) = \boldsymbol{z}^m$ we can compute the

**Proposition D.1** *Consider the GNN module in Fig. 1a with input $\boldsymbol{x}$ being a random vector with moments as in 3. The output of thes] GNN block in Fig. 1a, defined by 25, has a closed-form solution and is equivalent to 66, when $\sigma\left(\cdot\right) = \boldsymbol{z}^k$ and $\rho\left(\cdot\right) = \mathbb{E}\left[(\cdot) \odot \left(\mathbf{1}^T \boldsymbol{z}^{m-k}\right)\right]$, and equivalent to 67, when $\sigma\left(\boldsymbol{z}\right) = \boldsymbol{z}^k$ and $\rho\left(\cdot\right) = \mathbb{E}\left[(\cdot) \odot \left(\mathbf{1}^T \boldsymbol{z}\right)^{m-k}\right]$.*

We study the normalization function $\rho\left[\cdot\right] = (\cdot) \odot \mathbb{E}\left[\boldsymbol{z}^m\right]$ in the following section.

## E  MULTIPLE LAYER ANALYSIS

In the previous sections, we analyzed a single layer of 26, corresponding to multiple layers of 1. The type of outputs we get from this analysis are presented in 66 and 67, which are deterministic and can be given as input to other layers defined by either 1 or 26. In our analysis, the expectation operator enables a deterministic output that maintains permutation equivariance and is applied in the normalization layer following the first layer.

However, it can also be the case that $\mathbb{E}\left[\cdot\right]$ is applied after multiple layers. To understand the effect of multiple layers we study the following cases:

### E.1  TWO-LAYER GNN WITH $\sigma\left(\cdot\right) = (\cdot)^2$

We consider a two GNN layers defined by 26, where $\sigma\left(\cdot\right) = (\cdot)^k$, $\rho$ is equal to identity after the first layer and $\rho\left[\cdot\right] = \mathbb{E}\left[\cdot\right]$ after the second layer. Then the output of the first layer is:

$$\boldsymbol{x}^{(1)} = \boldsymbol{z}^2 = \left( \sum_{k=0}^{K} h_k^{(1)} \boldsymbol{S}^k \boldsymbol{x} \right)^2 \tag{68}$$

and the output of the second layer is:

$$\boldsymbol{x}^{(2)} = \mathbb{E}\left[\boldsymbol{w}^2\right] = \left( \sum_{k=0}^{K} h_k^{(2)} \boldsymbol{S}^k \boldsymbol{z}^2 \right)^2 \tag{69}$$

To derive an expression for $\boldsymbol{x}^{(2)}$ we study:

$$\mathbb{E}\left[\boldsymbol{w}\boldsymbol{w}^T\right] = \mathbb{E}\left[ \sum_{i=0}^{K} h_i^{(2)} \boldsymbol{S}^i \boldsymbol{z}^2 \boldsymbol{z}^{2^T} \sum_{j=0}^{K} h_j^{(2)} \boldsymbol{S}^j \right] = \sum_{i=0}^{K} h_i^{(2)} \boldsymbol{S}^i \mathbb{E}\left[\boldsymbol{z}^2 \boldsymbol{z}^{2^T}\right] \sum_{j=0}^{K} h_j^{(2)} \boldsymbol{S}^j. \tag{70}$$

To compute $\mathbb{E}\left[\boldsymbol{z}^2 \boldsymbol{z}^{2^T}\right]$ we use the expression in 62.

$$\mathbb{E}\left[\boldsymbol{w}\boldsymbol{w}^T\right] = \sum_{i=0}^{K} h_i^{(2)} \boldsymbol{S}^i \sum_{k,l,m,n=0}^{K} h_{k,l,m,n} \left( \boldsymbol{S}^k \odot \boldsymbol{S}^l \right) \left( \boldsymbol{S}^m \odot \boldsymbol{S}^n \right) \sum_{j=0}^{K} h_j^{(2)} \boldsymbol{S}^j$$

$$= \sum_{k,l,m,n,i,j=0}^{K} h_{k,l,m,n,i,j} \boldsymbol{S}^i \left( \boldsymbol{S}^k \odot \boldsymbol{S}^l \right) \left( \boldsymbol{S}^m \odot \boldsymbol{S}^n \right) \boldsymbol{S}^j, \tag{71}$$

and therefore:

$$\boldsymbol{x}^{(2)} = \sum_{k,l,m,n,i,j=0}^{K} h_{k,l,m,n,i,j} \text{diag}\left(\boldsymbol{S}^i\left(\boldsymbol{S}^k \odot \boldsymbol{S}^l\right)\left(\boldsymbol{S}^m \odot \boldsymbol{S}^n\right)\boldsymbol{S}^j\right), \tag{72}$$

### E.2 Two-layer GNN with $\rho\left[\cdot\right] = (\cdot) \odot \mathbb{E}\left[\mathbf{z}^2\right]$ in the first layer

Finally, we study a two-layer GNN defined by 26, where $\sigma$ is the identity in the first layer, $\sigma\left(\cdot\right) = (\cdot)^2$ in the second layer, $\rho\left[\cdot\right] = (\cdot) \odot \mathbb{E}\left[\mathbf{z}^2\right]$ after the first layer and $\rho\left[\cdot\right] = \mathbb{E}\left[\cdot\right]$ after the second layer. Then the output of the first layer is:

$$\boldsymbol{x}^{(1)} = \boldsymbol{z} \odot \mathbb{E}\left[\boldsymbol{z}^2\right] = \sum_{k=0}^{K} h_k^{(1)} \boldsymbol{S}^k \boldsymbol{x} \odot \mathbb{E}\left[\left(\sum_{k=0}^{K} h_k^{(1)} \boldsymbol{S}^k \boldsymbol{x}\right)^2\right] = \sum_{k=0}^{K} h_k^{(1)} \boldsymbol{S}^k \boldsymbol{x} \odot \sum_{k=0}^{K} \tilde{h}_k^{(1)} \text{diag}\left(\boldsymbol{S}^k\right)$$

$$= \sum_{k=0}^{K} \sum_{m=0}^{2K} h_k^{(1)} \tilde{h}_k^{(1)} \boldsymbol{S}^k \boldsymbol{x} \odot \left(\boldsymbol{S}^m \odot \boldsymbol{I}\right) = \sum_{k=0}^{K} \sum_{m=0}^{2K} h_k^{(1)} \tilde{h}_k^{(1)} \left(\boldsymbol{S}^m \odot \boldsymbol{I}\right) \boldsymbol{S}^k \boldsymbol{x} \tag{73}$$

The covariance of $\boldsymbol{x}^{(1)}$ takes the form:

$$\mathbb{E}\left[\boldsymbol{x}^{(1)} \boldsymbol{x}^{(1)^T}\right] = \mathbb{E}\left[\sum_{k=0}^{K} \sum_{m=0}^{2K} h_k^{(1)} \tilde{h}_k^{(1)} \left(\boldsymbol{S}^m \odot \boldsymbol{I}\right) \boldsymbol{S}^k \boldsymbol{x} \sum_{k=0}^{K} \sum_{m=0}^{2K} h_k^{(1)} \tilde{h}_k^{(1)} \boldsymbol{x}^T \boldsymbol{S}^k \left(\boldsymbol{S}^m \odot \boldsymbol{I}\right)\right]$$

$$= \sum_{k=0}^{K} \sum_{m=0}^{2K} h_k^{(1)} \tilde{h}_k^{(1)} \left(\boldsymbol{S}^m \odot \boldsymbol{I}\right) \boldsymbol{S}^k \mathbb{E}\left[\boldsymbol{x}\boldsymbol{x}^T\right] \sum_{k=0}^{K} \sum_{m=0}^{2K} h_k^{(1)} \tilde{h}_k^{(1)} \boldsymbol{S}^k \left(\boldsymbol{S}^m \odot \boldsymbol{I}\right)$$

$$= \sum_{k,l,m} h'_{k,l,m} \left(\boldsymbol{S}^k \odot \boldsymbol{I}\right) \boldsymbol{S}^l \left(\boldsymbol{S}^m \odot \boldsymbol{I}\right) \tag{74}$$

and the output of the second layer is:

$$\boldsymbol{x}^{(2)} = \mathbb{E}\left[\boldsymbol{w}^2\right] = \left(\sum_{k=0}^{K} h_k^{(2)} \boldsymbol{S}^k \boldsymbol{x}^{(1)}\right)^2 \tag{75}$$

To derive an expression for $\boldsymbol{x}^{(2)}$ we study:

$$\mathbb{E}\left[\boldsymbol{w}\boldsymbol{w}^T\right] = \mathbb{E}\left[\sum_{i=0}^{K} h_i^{(2)} \boldsymbol{S}^i \boldsymbol{x}^{(1)} \boldsymbol{x}^{(1)^T} \sum_{j=0}^{K} h_j^{(2)} \boldsymbol{S}^j\right] = \sum_{i=0}^{K} h_i^{(2)} \boldsymbol{S}^i \mathbb{E}\left[\boldsymbol{x}^{(1)} \boldsymbol{x}^{(1)^T}\right] \sum_{j=0}^{K} h_j^{(2)} \boldsymbol{S}^j$$

$$= \sum_{k,l,m,i,j=0}^{K} h_{k,l,m,i,j} \boldsymbol{S}^i \left(\boldsymbol{S}^k \odot \boldsymbol{I}\right) \boldsymbol{S}^l \left(\boldsymbol{S}^m \odot \boldsymbol{I}\right) \boldsymbol{S}^j. \tag{76}$$

As a result:

$$\boldsymbol{x}^{(2)} = \sum_{k,l,m,i,j=0}^{K} h_{k,l,m,i,j} \text{diag}\left(\boldsymbol{S}^i \left(\boldsymbol{S}^k \odot \boldsymbol{I}\right) \boldsymbol{S}^l \left(\boldsymbol{S}^m \odot \boldsymbol{I}\right) \boldsymbol{S}^j\right). \tag{77}$$

We also study a two-layer GNN defined by 26, where $\sigma$ is the identity in the first layer and the second layer, $\rho\left[\cdot\right] = (\cdot) \odot \mathbb{E}\left[\boldsymbol{z}^2\right]$ after the first layer and $\rho\left(\cdot\right) = \mathbb{E}\left[(\cdot) \odot \left(\mathbf{1}^T \boldsymbol{z}\right)\right]$ after the second layer. Then the output of the second layer is:

$$\boldsymbol{x}^{(2)} = \mathbb{E}\left[\boldsymbol{w}\left(\mathbf{1}^T \boldsymbol{w}\right)\right] = \mathbb{E}\left[\boldsymbol{w}\boldsymbol{w}^T\right]\mathbf{1} \tag{78}$$

As a result:

$$\boldsymbol{x}^{(2)} = \sum_{k,l,m,i,j=0}^{K} h_{k,l,m,i,j} \boldsymbol{S}^i \left(\boldsymbol{S}^m \odot \boldsymbol{I}\right) \boldsymbol{S}^l \left(\boldsymbol{S}^m \odot \boldsymbol{I}\right) \boldsymbol{S}^j \mathbf{1}. \tag{79}$$

## F   COUNTING CONNECTED COMPONENTS: PROOF OF THEOREM 4.1

To prove Theorem 4.1, consider the GNN definition in 4, where we use the Laplacian of a graph as the graph operator $\boldsymbol{S}$. Then the spectral decomposition of $\boldsymbol{S}$ is:

$$\boldsymbol{S} = \boldsymbol{U}\boldsymbol{\Lambda}\boldsymbol{U}^T = \sum_n \lambda_n \boldsymbol{u}_n \boldsymbol{u}_n^T, \tag{80}$$

where $\lambda_n$ is an eigenvalue of $\boldsymbol{S}$ with corresponding eigenvector $\boldsymbol{u}_n$. The graph convolution in 4 can be cast as:

$$\boldsymbol{z} = \sum_{k=0}^K h_k \boldsymbol{S}^k \boldsymbol{x} = \sum_{k=0}^K h_k \boldsymbol{U}\boldsymbol{\Lambda}^k \boldsymbol{U}^T \boldsymbol{x} = \sum_{k=0}^K h_k \sum_n \lambda_n{}^k \boldsymbol{u}_n \boldsymbol{u}_n^T \boldsymbol{x}$$

$$= \sum_n \sum_{k=0}^K h_k \lambda_n^k \boldsymbol{u}_n \boldsymbol{u}_n^T \boldsymbol{x} = \sum_n \bar{\boldsymbol{h}}\left(\lambda_n\right) \boldsymbol{u}_n \boldsymbol{u}_n^T \boldsymbol{x}, \tag{81}$$

where

$$\bar{\boldsymbol{h}}\left(\lambda_n\right) = \sum_{k=0}^K h_k \lambda_n^k, \tag{82}$$

is the frequency representation of the graph filter. Now let $\{\mu_1, \ldots, \mu_q\}$ be the set of the distinct eigenvalues of a set of Laplacians $\{\boldsymbol{S}_i\}_{i=1}^I$ corresponding to a set of graphs $\{\mathcal{G}_i\}_{i=1}^I$. Then we can write the following linear system of equations:

$$\begin{bmatrix} \bar{\boldsymbol{h}}\left(\mu_1\right) \\ \bar{\boldsymbol{h}}\left(\mu_2\right) \\ \vdots \\ \bar{\boldsymbol{h}}\left(\mu_q\right) \end{bmatrix} = \begin{bmatrix} 1\ \mu_1\ \mu_1^2 \ldots \mu_1^{K-1} \\ 1\ \mu_2\ \mu_2^2 \ldots \mu_2^{K-1} \\ \vdots \\ 1\ \mu_q\ \mu_q^2 \ldots \mu_q^K \end{bmatrix} \begin{bmatrix} h_0 \\ h_1 \\ \vdots \\ h_{K-1} \end{bmatrix} = \boldsymbol{W}\boldsymbol{h} \tag{83}$$

$\boldsymbol{W}$ is a Vandermonde matrix and when $K = q - 1$ the determinant of $\boldsymbol{W}$ takes the form:

$$\det\left(\boldsymbol{W}\right) = \boldsymbol{\Pi}_{1 \le i < j \le q}\left(\mu_i - \mu_j\right) \tag{84}$$

Since the values $\mu_i$ are distinct, $\boldsymbol{W}$ has full column rank and there exists a graph filter with unique parameters $\boldsymbol{h}$ that passes only one eigenvalue (the $k$-th eigenvalue), i.e.,

$$\bar{\boldsymbol{h}}\left(\mu_i\right) = \begin{cases} 1, & \text{if } i = k \\ 0, & \text{if } i \ne k \end{cases} \tag{85}$$

The number of connected components in a graph is equal to the multiplicity of the zero eigenvalue of the Laplacian matrix. We can design a graph filter to isolate the 0 eigenvalue i.e.,

$$\bar{\boldsymbol{h}}\left(\lambda\right) = \begin{cases} 1, & \text{if } \lambda = 0 \\ 0, & \text{if } \lambda \ne 0 \end{cases} \tag{86}$$

If we use the graph filter in 86, in the definition of graph convolution in 81 we get:

$$\boldsymbol{z} = \boldsymbol{U}_0 \boldsymbol{U}_0^T \boldsymbol{x}, \tag{87}$$

where $\boldsymbol{U}_0$ is the eigenspace corresponding to the zero eigenvalue of the Laplacian. Then we feed 87 with a random input $\boldsymbol{x} \in \mathbb{R}^N$ with characteristic function $\phi_{\boldsymbol{x}}\left(t_1, \ldots, t_N\right) = \prod_{i=1}^N \left(e^{jt_i} - jt_i\right)$ and moments defined in 3. If $\sigma$ is the elementwise square function and $\rho$ is the expectation operator, the output in 87 takes the yields:

$$\boldsymbol{y} = \mathbb{E}\left[\boldsymbol{z}^2\right] = \text{diag}\left(\mathbb{E}\left[\boldsymbol{z}\boldsymbol{z}^T\right]\right) = \text{diag}\left(\sum_{k=0}^{K-1} h_k \boldsymbol{S}^k \mathbb{E}\left[\boldsymbol{x}\boldsymbol{x}^T\right] \sum_{m=0}^{K-1} h_m \boldsymbol{S}^m\right)$$

$$= \text{diag}\left(\sum_{k=0}^{K-1} h_k \boldsymbol{S}^k \sum_{m=0}^{K-1} h_m \boldsymbol{S}^m\right) = \text{diag}\left(\boldsymbol{U}_0 \boldsymbol{U}_0^T \boldsymbol{U}_0 \boldsymbol{U}_0^T\right) = \text{diag}\left(\boldsymbol{U}_0 \boldsymbol{U}_0^T\right). \tag{88}$$

Therefore the output $\boldsymbol{y}$ is the diagonal of a matrix with a rank equal to the number of connected components $r$. The eigenvalues of $\boldsymbol{U}_0\boldsymbol{U}_0^T$ are $\lambda = 1$ with multiplicity equal to the number of connected components $r$ and $\lambda = 0$ with multiplicity $N - r$. Then $\mathbf{1}^T\boldsymbol{y}$ takes the form:

$$\mathbf{1}^T\boldsymbol{y} = \text{trace}\left(\boldsymbol{U}_0\boldsymbol{U}_0^T\right) = \sum_n \lambda_n = r. \tag{89}$$

and a single layer GNN, defined in 87 can count the number of connected components of graphs $\{\mathcal{G}_i\}_{i=1}^I$.

## G   COUNTING CYCLES

To prove that the considered message-passing GNNs can count cycles, we use the formulas in Perepechko & Voropaev (2009) that associate the number of cycles in a graph with the graph adjacency.

### G.1   PROOF OF THEOREM 4.2:

The number of triangles within a graph is defined as follows:

$$c^3 = \frac{1}{6}\mathbf{1}^T\left(\boldsymbol{S}^2 \odot \boldsymbol{S}\right)\mathbf{1} \tag{90}$$

This can be computed by the expression in 6. The number of tetragons and pentagons within a graph is defined as follows:

$$c^4 = \frac{1}{8}\mathbf{1}^T\left(\left(\boldsymbol{S}^2 \odot \boldsymbol{S}^2\right)\mathbf{1} + \left(\boldsymbol{S}^2 \odot \boldsymbol{I}\right)\mathbf{1} - 2\boldsymbol{S}\left(\boldsymbol{S}^2 \odot \boldsymbol{I}\right)\mathbf{1}\right) \tag{91}$$

$$c^5 = \frac{1}{10}\mathbf{1}^T\left(\left(\boldsymbol{S}^3 \odot \boldsymbol{S}^2\right)\mathbf{1} + 5\left(\boldsymbol{S}^2 \odot \boldsymbol{S}\right)\mathbf{1} - 5\boldsymbol{S}\left(\boldsymbol{S}^2 \odot \boldsymbol{S}\right)\mathbf{1}\right) \tag{92}$$

All terms in $c_4$, $c_5$ can be computed by 6, apart from $\boldsymbol{S}\left(\boldsymbol{S}^2 \odot \boldsymbol{I}\right)$ and $\boldsymbol{S}\left(\boldsymbol{S}^2 \odot \boldsymbol{S}\right)$. To compute these terms we just need to pass 6 through one more MPNN layer.

### G.2   PROOF OF THEOREM 4.3:

$$\begin{aligned}
c^6 = \frac{1}{12}\mathbf{1}^T\Big( & \left(\boldsymbol{S}^3 \odot \boldsymbol{S}^3\right)\mathbf{1} - 3\left(\boldsymbol{S}^3 \odot \boldsymbol{S}^3 \odot \boldsymbol{I}\right)\mathbf{1} + 9\left(\boldsymbol{S}^2 \odot \boldsymbol{S}^2 \odot \boldsymbol{S}\right)\mathbf{1} - 6\boldsymbol{S}\left(\boldsymbol{S}^2 \odot \boldsymbol{S}^2\right)\mathbf{1} \\
& + 6\left(\boldsymbol{S}^2 \odot \boldsymbol{S}^2\right)\mathbf{1} - 4\left(\boldsymbol{S}^2 \odot \boldsymbol{S}\right)\mathbf{1} + 4\boldsymbol{S}\left(\boldsymbol{S}^2 \odot \boldsymbol{S}^2 \odot \boldsymbol{I}\right)\mathbf{1} + 3\boldsymbol{S}^2\left(\boldsymbol{S}^2 \odot \boldsymbol{I}\right)\mathbf{1} \\
& - 12\boldsymbol{S}\left(\boldsymbol{S}^2 \odot \boldsymbol{I}\right)\mathbf{1} + 4\left(\boldsymbol{S}^2 \odot \boldsymbol{I}\right)\mathbf{1}\Big)
\end{aligned} \tag{93}$$

We observe 5 types of terms in $c_6$:

$$\left(\boldsymbol{S}^k \odot \boldsymbol{S}^l\right)\mathbf{1} \tag{94}$$
$$\boldsymbol{S}\left(\boldsymbol{S}^k \odot \boldsymbol{S}^l\right)\mathbf{1} \tag{95}$$
$$\boldsymbol{S}^2\left(\boldsymbol{S}^k \odot \boldsymbol{S}^l\right)\mathbf{1} \tag{96}$$
$$\left(\boldsymbol{S}^k \odot \boldsymbol{S}^l \odot \boldsymbol{S}^m\right)\mathbf{1} \tag{97}$$
$$\boldsymbol{S}\left(\boldsymbol{S}^k \odot \boldsymbol{S}^l \odot \boldsymbol{S}^m\right)\mathbf{1} \tag{98}$$

The term in 94 can be computed by 6, whereas 95 and 96 are computed by passing 6 through one or two more MPNN layers respectively. Similarly the term in 97 can be computed by 10, whereas 98 by passing 10 through one more MPNN layer.

### G.3 PROOF OF THEOREM 4.5:

$$c^7 = \frac{1}{14}\mathbf{1}^T\Big(\left(\boldsymbol{S}^4 \odot \boldsymbol{S}^3\right)\mathbf{1} - 7\left(\boldsymbol{S}^4 \odot \boldsymbol{S}^3 \odot \boldsymbol{I}\right)\mathbf{1} + 7\left(\boldsymbol{S}^2 \odot \boldsymbol{S}^2 \odot \boldsymbol{S}^2 \odot \boldsymbol{S}\right)\mathbf{1}$$
$$- 7\left(\boldsymbol{S}^5 \odot \boldsymbol{S}^2 \odot \boldsymbol{I}\right)\mathbf{1} + 21\left(\boldsymbol{S}^3 \odot \boldsymbol{S}^2 \odot \boldsymbol{S}\right)\mathbf{1} - 28\left(\boldsymbol{S}^2 \odot \boldsymbol{S}^2 \odot \boldsymbol{S}\right)\mathbf{1}$$
$$+ 14\left(\boldsymbol{S}^3 \odot \boldsymbol{S}^2 \odot \boldsymbol{S}^2 \odot \boldsymbol{I}\right)\mathbf{1} + 7\left(\boldsymbol{S}^3 \odot \boldsymbol{S}^2\right)\mathbf{1} + 7\boldsymbol{S}^2\left(\boldsymbol{S}^2 \odot \boldsymbol{S}\right)\mathbf{1}$$
$$+ 7\boldsymbol{S}\left(\boldsymbol{S}^2 \odot \boldsymbol{S}\right)\boldsymbol{S}\mathbf{1} - 77\left(\boldsymbol{S}^2 \odot \boldsymbol{S}^3 \odot \boldsymbol{I}\right)\mathbf{1} + 56\left(\boldsymbol{S}^2 \odot \boldsymbol{S}\right)\mathbf{1}\Big) \tag{99}$$

We observe 5 types of terms in $c_7$:

$$\left(\boldsymbol{S}^k \odot \boldsymbol{S}^l\right)\mathbf{1} \tag{100}$$
$$\boldsymbol{S}^2\left(\boldsymbol{S}^k \odot \boldsymbol{S}^l\right)\mathbf{1} \tag{101}$$
$$\boldsymbol{S}\left(\boldsymbol{S}^k \odot \boldsymbol{S}^l\right)\boldsymbol{S}\mathbf{1} \tag{102}$$
$$\left(\boldsymbol{S}^k \odot \boldsymbol{S}^l \odot \boldsymbol{S}^m\right)\mathbf{1} \tag{103}$$
$$\left(\boldsymbol{S}^k \odot \boldsymbol{S}^l \odot \boldsymbol{S}^m \odot \boldsymbol{S}^n\right)\mathbf{1} \tag{104}$$

The terms in 100, 103, and 104 can be computed by 6, 10, and 13 respectively. The term in 101 can be computed by passing 6 through 2 MPNN layers, whereas the term in 102 can be computed by passing 12 through 1 MPNN layer.

## H COUNTING QUASI-CLIQUES

To prove that the considered message-passing GNNs can count quasi-cliques, we use the formulas in Barik & Reddy (2023) that associate the number of certain substructures in a graph with the graph adjacency.

### H.1 PROOF OF THEOREM 4.4

The authors in Barik & Reddy (2023) derive the following equations:

$$Q_4 = \frac{1}{8}\mathbf{1}^T\left(\boldsymbol{S}^2 \odot \boldsymbol{S}^2\right)\mathbf{1} - \frac{1}{8}\mathbf{1}^T\left(\boldsymbol{S}^2 \odot \boldsymbol{S}^2\boldsymbol{I}\right)\mathbf{1} + \frac{1}{8}\mathbf{1}^T\left(\boldsymbol{S}^2 \odot \boldsymbol{I}\odot\right)\mathbf{1} - \frac{1}{8}\mathbf{1}^T\boldsymbol{S}^2\mathbf{1} \tag{105}$$
$$Q_5 = \frac{1}{2}\mathbf{1}^T\left(\boldsymbol{S}^3 \odot \boldsymbol{S}^2 \odot \boldsymbol{S}\right)\mathbf{1} + \frac{1}{2}\mathbf{1}^T\left(\boldsymbol{S}^2 \odot \boldsymbol{S}\right)\mathbf{1} - \mathbf{1}^T\boldsymbol{S}\left(\boldsymbol{S}^2 \odot \boldsymbol{S}\right)\mathbf{1} - 4Q_4, \tag{106}$$

which can be computed by 6, and 10 followed by a classical message-passing layer.

## I PROOF OF PROPOSITION 5.2

The proposed architecture is more powerful, for certain problem classes, than the 1-FWL test (1-WL and 2-WL) since it can count graph structures that 1-FWL cannot. It is strictly more powerful since it can produce exactly the same outputs as 1-FWL when $f$, $g$ are injective.

## J PROOF OF THEOREMS 6.1, 6.2, 6.3

### J.1 LISTING ALL THE TRIANGLES WITHIN A GRAPH

Now let $\underline{\boldsymbol{T}}$ denote the triangle tensor, i.e.,

$$\boldsymbol{T}[i,j,k] = \boldsymbol{S}[i,j]\ \boldsymbol{S}[j,k]\ \boldsymbol{S}[i,k]. \tag{107}$$

$\underline{\boldsymbol{T}}[i,j,k] = 1$ if nodes $i,j,k$ form a triangle and $\underline{\boldsymbol{T}}[i,j,k] = 0$ otherwise. Unfortunately, $\underline{\boldsymbol{T}}$ does not admit an obvious tensor model with factor matrices the adjacency powers.

Instead, we will consider a different type of tensor $\underline{\boldsymbol{G}}$ such that:

$$\underline{\boldsymbol{G}}[i,j,k] = \boldsymbol{S}\left[i,k\right]\boldsymbol{S}\left[j,k\right] + \boldsymbol{S}\left[i,j\right]\boldsymbol{S}\left[j,k\right] + \boldsymbol{S}\left[i,k\right]\boldsymbol{S}\left[i,j\right], \tag{108}$$

which is a simplician complex tensor that measures relationships between node triplets. In particular $\boldsymbol{G}[i,j,k] = 3$ when $(i,j,k)$, is a triangle, $\boldsymbol{G}[i,j,k] = 1$ when there are two edges between $i, j, k$ or when $i = j$ and $(i,k) \in \mathcal{E}$. $\boldsymbol{G}[i,j,k] = 1$ also when $i = k$ and $(i,j) \in \mathcal{E}$ or when $j = k$ and $(i,k) \in \mathcal{E}$. $\boldsymbol{G}[i,j,k] = 0$ in any other case. It is therefore clear that:

$$\underline{\boldsymbol{T}} = \mathrm{mod}_3\left[\underline{\boldsymbol{G}}\right] = \mathtt{ReLU}\left[\underline{\boldsymbol{G}} - 2\right] \tag{109}$$

Interestingly $\underline{\boldsymbol{G}}$ can be written using the following CPD models:

$$\underline{\boldsymbol{G}} = [\![\boldsymbol{S}, \boldsymbol{S}, \boldsymbol{I}]\!] + [\![\boldsymbol{S}, \boldsymbol{I}, \boldsymbol{S}]\!] + [\![\boldsymbol{I}, \boldsymbol{S}, \boldsymbol{S}]\!] \tag{110}$$

From equation 8 we get that:

$$\mathbb{E}\left[\boldsymbol{z} \circ \boldsymbol{z} \circ \boldsymbol{z}\right] = [\![\boldsymbol{H}\left(\boldsymbol{S}\right), \boldsymbol{H}\left(\boldsymbol{S}\right), \boldsymbol{H}\left(\boldsymbol{S}\right)]\!] = \sum_{i_1,\dots,i_3=0}^{K} h_{i_1,i_2,i_3}\left[\![\boldsymbol{S}^{i_1}, \boldsymbol{S}^{i_2}, \boldsymbol{S}^{i_3}]\!\right], \tag{111}$$

which concludes our proof.

### J.2 Counting cliques

The tensor of $4$-th order cliques is:

$$\underline{\boldsymbol{Q}}_4[i,j,k,l] = \boldsymbol{S}\left[i,j\right]\boldsymbol{S}\left[j,k\right]\boldsymbol{S}\left[i,k\right]\boldsymbol{S}\left[i,l\right]\boldsymbol{S}\left[j,l\right]\boldsymbol{S}\left[k,l\right]. \tag{112}$$

$\underline{\boldsymbol{Q}}_4[i,j,k,l] = 1$ if nodes $i, j, k, l$ form a clique and $\underline{\boldsymbol{Q}}_4[i,j,k,l] = 0$ otherwise. Unfortunately $\underline{\boldsymbol{Q}}_4$ does not admit an obvious tensor model with factor matrices the adjacency powers. The number of $4$-node cliques in a graph is given by:

$$q_4 = \sum_{i,j,k,l} \boldsymbol{S}\left[i,j\right]\boldsymbol{S}\left[j,k\right]\boldsymbol{S}\left[i,k\right]\boldsymbol{S}\left[i,l\right]\boldsymbol{S}\left[j,l\right]\boldsymbol{S}\left[k,l\right], \tag{113}$$

which is equal to:

$$q_4 = \sum_{i,j,k,l} \boldsymbol{T}[i,j,k]\boldsymbol{S}\left[i,l\right]\boldsymbol{S}\left[j,l\right]\boldsymbol{S}\left[k,l\right]. \tag{114}$$

By rearranging the summations we get:

$$q_4 = \sum_{l}\sum_{i,j,k} \boldsymbol{T}[i,j,k]\boldsymbol{S}\left[i,l\right]\boldsymbol{S}\left[j,l\right]\boldsymbol{S}\left[k,l\right]. \tag{115}$$

The inner summation corresponds to the diagonal element of a third-order tensor with Tucker model:

$$[\![\underline{\boldsymbol{T}}, \boldsymbol{S}, \boldsymbol{S}, \boldsymbol{S}]\!], \tag{116}$$

where $\underline{\boldsymbol{T}}$ is the core tensor and $\boldsymbol{S}$ are the factor matrices. To compute the number of $4$-node cliques we nee to sum the superdiagonal of the tensor above, i.e.,

$$q_4 = \boldsymbol{1}^T \mathrm{superdiag}\left([\![\underline{\boldsymbol{T}}, \boldsymbol{S}, \boldsymbol{S}, \boldsymbol{S}]\!]\right). \tag{117}$$

When the input to the GNN is $\boldsymbol{x}_t$ as described in Section 6 then

$$\begin{aligned}
\mathbb{E}\left[\boldsymbol{z}^3\right] &= \mathrm{superdiag}\left([\![\underline{\boldsymbol{T}}, \boldsymbol{H}\left(\boldsymbol{S}\right), \boldsymbol{H}\left(\boldsymbol{S}\right), \boldsymbol{H}\left(\boldsymbol{S}\right)]\!]\right) \\
&= \sum_{i_1,\dots,i_3=0}^{K} h_{i_1,i_2,i_3}\mathrm{superdiag}\left([\![\underline{\boldsymbol{T}}, \boldsymbol{S}^{i_1}, \boldsymbol{S}^{i_2}, \boldsymbol{S}^{i_3}]\!]\right),
\end{aligned} \tag{118}$$

which concludes our proof.

## J.3 COUNTING 8-NODE CYCLES

To count the number of 8-node cycles in a graph using message-passing operations we use Lemma 7 in Barik & Reddy (2023). All the terms in this formula can be computed using Hadamard product operations, as discussed in other sections. The only terms that cannot be computed with Hadamard products are those of the following form:

$$\underline{\boldsymbol{Q}}_4[i,j,k,l] = \boldsymbol{S}[i,j] \ \boldsymbol{S}[j,k] \ \boldsymbol{S}[i,k] \boldsymbol{S}^{i_1}[i,l] \ \boldsymbol{S}^{i_2}[j,l] \ \boldsymbol{S}^{i_3}[k,l]. \tag{119}$$

This term can be computed by the procedure discussed in Section 6 and J.2, i.e., by equation 118. This concludes our proof.

## K COUNTING CYCLES AT THE NODE LEVEL

In this section, we study the ability of GNNs defined in 1 or 26 to produce permutation equivariant node representations that can count the number of cycles each node is involved in. To this end, we derive the following theorem.

**Theorem K.1 (Node-level Cycles)** *There exists a GNN defined in 1 or 26 that generates permutation equivariant node representations that count the number of $k-$size cycles each node participates, for any graph and $k \leq 7$.*

### K.1 PROOF OF THEOREM K.1

To prove Theorem K.1 we use the expressions in Perepechko & Voropaev (2009) that instantiate the matrices of $k-$paths $\boldsymbol{P}_k$. To be more precise $\boldsymbol{P}_k(i,j)$ counts the number of paths $k-$paths between nodes $i$ and $j$.

### K.2 MATRICES OF $k-$PATHS FOR UNDIRECTED GRAPHS

Starting from the definition of $k-$path matrices, as presented in Perepechko & Voropaev (2009), we can compute $k-$path matrices $\boldsymbol{P}_k$ for undirected graphs. After some algebraic manipulations $\boldsymbol{P}_k$ for $3 \leq k \leq 6$ can be cast as:

$$\boldsymbol{P}_3 = \boldsymbol{S}^3 - \left(\boldsymbol{S}^2 \odot \boldsymbol{I}\right) \boldsymbol{S} - \boldsymbol{S}\left(\boldsymbol{S}^2 \odot \boldsymbol{I}\right) + \boldsymbol{S} \tag{120}$$

$$\begin{aligned} \boldsymbol{P}_4 =& \boldsymbol{S}^4 - \left(\boldsymbol{S}^3 \odot \boldsymbol{I}\right) \boldsymbol{S} - \boldsymbol{S}\left(\boldsymbol{S}^3 \odot \boldsymbol{I}\right) - \left(\boldsymbol{S}^2 \odot \boldsymbol{I}\right) \boldsymbol{S}^2 - \boldsymbol{S}^2 \left(\boldsymbol{S}^2 \odot \boldsymbol{I}\right) \\ & - \boldsymbol{S}\left(\boldsymbol{S}^2 \odot \boldsymbol{I}\right) \boldsymbol{S} + 3 \left(\boldsymbol{S} \odot \boldsymbol{S}^2\right) + 2\boldsymbol{S}^2 \end{aligned} \tag{121}$$

$$\begin{aligned} \boldsymbol{P}_5 =& \boldsymbol{S}^5 - \left(\boldsymbol{S}^4 \odot \boldsymbol{I}\right) \boldsymbol{S} - \boldsymbol{S}\left(\boldsymbol{S}^4 \odot \boldsymbol{I}\right) - \left(\boldsymbol{S}^3 \odot \boldsymbol{I}\right) \boldsymbol{S}^2 - \boldsymbol{S}^2 \left(\boldsymbol{S}^3 \odot \boldsymbol{I}\right) \\ & - \left(\boldsymbol{S}^2 \odot \boldsymbol{I}\right) \boldsymbol{S}^3 - \boldsymbol{S}^3 \left(\boldsymbol{S}^2 \odot \boldsymbol{I}\right) + 3 \left(\boldsymbol{S}^3 \odot \boldsymbol{S}\right) - \boldsymbol{S}\left(\boldsymbol{S}^3 \odot \boldsymbol{I}\right) \boldsymbol{S} \\ & + 2 \left(\boldsymbol{S}^2 \odot \boldsymbol{S}^2 \odot \boldsymbol{I}\right) \boldsymbol{S} + 2\boldsymbol{S}\left(\boldsymbol{S}^2 \odot \boldsymbol{S}^2 \odot \boldsymbol{I}\right) + 3 \left(\boldsymbol{S}^2 \odot \boldsymbol{S}^2 \odot \boldsymbol{S}\right) \\ & + \left(\boldsymbol{S}^2 \odot \boldsymbol{I}\right) \boldsymbol{S} \left(\boldsymbol{S}^2 \odot \boldsymbol{I}\right) - \boldsymbol{S}\left(\boldsymbol{S}^2 \odot \boldsymbol{I}\right) \boldsymbol{S}^2 \\ & - \boldsymbol{S}^2 \left(\boldsymbol{S}^2 \odot \boldsymbol{I}\right) \boldsymbol{S} + 3 \left(\boldsymbol{S}^2 \odot \boldsymbol{S}\right) \boldsymbol{S} + 3\boldsymbol{S}\left(\boldsymbol{S}^2 \odot \boldsymbol{S}\right) \\ & + \left(\left(\boldsymbol{S}\left(\boldsymbol{S}^2 \odot \boldsymbol{I}\right) \boldsymbol{S}\right) \odot \boldsymbol{I}\right) \boldsymbol{S} + \boldsymbol{S}\left(\left(\boldsymbol{S}\left(\boldsymbol{S}^2 \odot \boldsymbol{I}\right) \boldsymbol{S}\right) \odot \boldsymbol{I}\right) \\ & - 4 \left(\boldsymbol{S} \odot \boldsymbol{S}^2\right) - 6 \left(\boldsymbol{I} \odot \boldsymbol{S}^2\right) \boldsymbol{S} - 6\boldsymbol{S}\left(\boldsymbol{I} \odot \boldsymbol{S}^2\right) \\ & + 3\boldsymbol{S}^3 + 4\boldsymbol{S} \end{aligned} \tag{122}$$

The number of terms in $\boldsymbol{P}_6$ is large enough and therefore we split them in 4 terms, i.e., $\boldsymbol{P}_6 = \boldsymbol{P}_6^1 + \boldsymbol{P}_6^2 + \boldsymbol{P}_6^3 + \boldsymbol{P}_6^4$.

$$
\begin{aligned}
\boldsymbol{P}_6^1 = & \boldsymbol{S}^6 - \left(\boldsymbol{S}^5 \odot \boldsymbol{I}\right)\boldsymbol{S} - \boldsymbol{S}\left(\boldsymbol{S}^5 \odot \boldsymbol{I}\right) - \left(\boldsymbol{S}^2 \odot \boldsymbol{I}\right)\boldsymbol{S}^4 - \boldsymbol{S}^4\left(\boldsymbol{S}^2 \odot \boldsymbol{I}\right) \\
& - \left(\boldsymbol{S}^4 \odot \boldsymbol{I}\right)\boldsymbol{S}^2 - \boldsymbol{S}^2\left(\boldsymbol{S}^4 \odot \boldsymbol{I}\right) - \boldsymbol{S}\left(\boldsymbol{S}^4 \odot \boldsymbol{I}\right)\boldsymbol{S} + 2\left(\boldsymbol{S}^4 \odot \boldsymbol{S}\right) \\
& - \left(\boldsymbol{S}^3 \odot \boldsymbol{I}\right)\boldsymbol{S}^3 - \boldsymbol{S}^3\left(\boldsymbol{S}^3 \odot \boldsymbol{I}\right) - \boldsymbol{S}\left(\boldsymbol{S}^2 \odot \boldsymbol{I}\right)\boldsymbol{S}^3 - \boldsymbol{S}^3\left(\boldsymbol{S}^2 \odot \boldsymbol{I}\right)\boldsymbol{S} \\
& - \boldsymbol{S}\left(\boldsymbol{S}^3 \odot \boldsymbol{I}\right)\boldsymbol{S}^2 - \boldsymbol{S}^2\left(\boldsymbol{S}^3 \odot \boldsymbol{I}\right)\boldsymbol{S} + 4\left(\boldsymbol{S}^3 \odot \boldsymbol{S}^2 \odot \boldsymbol{I}\right)\boldsymbol{S} + 4\boldsymbol{S}\left(\boldsymbol{S}^3 \odot \boldsymbol{S}^2 \odot \boldsymbol{I}\right) \\
& + 6\left(\boldsymbol{S}^3 \odot \boldsymbol{S}^2 \odot \boldsymbol{S}\right) + \left(\boldsymbol{S}^2 \odot \boldsymbol{I}\right)\boldsymbol{S}\left(\boldsymbol{S}^3 \odot \boldsymbol{I}\right) + \left(\boldsymbol{S}^3 \odot \boldsymbol{I}\right)\boldsymbol{S}\left(\boldsymbol{S}^2 \odot \boldsymbol{I}\right) \\
& + 2\left(\boldsymbol{S}^3 \odot \boldsymbol{S}\right)\boldsymbol{S} + 2\boldsymbol{S}\left(\boldsymbol{S}^3 \odot \boldsymbol{S}\right) \\
& + \left(\left(\boldsymbol{S}\left(\boldsymbol{S}^3 \odot \boldsymbol{I}\right)\boldsymbol{S}\right) \odot \boldsymbol{I}\right)\boldsymbol{S} + \boldsymbol{S}\left(\left(\boldsymbol{S}\left(\boldsymbol{S}^3 \odot \boldsymbol{I}\right)\boldsymbol{S}\right) \odot \boldsymbol{I}\right) \\
& - \boldsymbol{S}^2\left(\boldsymbol{S}^2 \odot \boldsymbol{I}\right)\boldsymbol{S}^2 + 2\left(\boldsymbol{S}^2 \odot \boldsymbol{S}^2 \odot \boldsymbol{I}\right)\boldsymbol{S}^2 + 2\boldsymbol{S}^2\left(\boldsymbol{S}^2 \odot \boldsymbol{S}^2 \odot \boldsymbol{I}\right) \\
& + \left(\boldsymbol{S}^2 \odot \boldsymbol{S}^2 \odot \boldsymbol{S}^2\right) + \left(\boldsymbol{S}^2 \odot \boldsymbol{I}\right)\boldsymbol{S}^2\left(\boldsymbol{S}^2 \odot \boldsymbol{I}\right) \quad (123)
\end{aligned}
$$

$$
\begin{aligned}
\boldsymbol{P}_6^2 = & 2\left(\boldsymbol{S}^2 \odot \boldsymbol{S}\right)\boldsymbol{S}^2 + 2\boldsymbol{S}^2\left(\boldsymbol{S}^2 \odot \boldsymbol{S}\right) \\
& + \left(\left(\boldsymbol{S}\left(\boldsymbol{S}^2 \odot \boldsymbol{I}\right)\boldsymbol{S}\right) \odot \boldsymbol{I}\right)\boldsymbol{S}^2 + \boldsymbol{S}^2\left(\left(\boldsymbol{S}\left(\boldsymbol{S}^2 \odot \boldsymbol{I}\right)\boldsymbol{S}\right) \odot \boldsymbol{I}\right) \\
& + \left(\boldsymbol{S}^2 \odot \boldsymbol{I}\right)\boldsymbol{S}\left(\boldsymbol{S}^2 \odot \boldsymbol{I}\right)\boldsymbol{S} + \boldsymbol{S}\left(\boldsymbol{S}^2 \odot \boldsymbol{I}\right)\boldsymbol{S}\left(\boldsymbol{S}^2 \odot \boldsymbol{I}\right) \\
& + \left(\left(\boldsymbol{S}\left(\boldsymbol{S}^2 \odot \boldsymbol{I}\right)\boldsymbol{S}^2\right) \odot \boldsymbol{I}\right)\boldsymbol{S} + \boldsymbol{S}\left(\left(\boldsymbol{S}^2\left(\boldsymbol{S}^2 \odot \boldsymbol{I}\right)\boldsymbol{S}\right) \odot \boldsymbol{I}\right) \\
& + 2\left(\boldsymbol{S} \odot \boldsymbol{S}^2 \odot \boldsymbol{S}^2\right)\boldsymbol{S} + 2\boldsymbol{S}\left(\boldsymbol{S} \odot \boldsymbol{S}^2 \odot \boldsymbol{S}^2\right) \\
& + \left(\left(\boldsymbol{S}^2\left(\boldsymbol{S}^2 \odot \boldsymbol{I}\right)\boldsymbol{S}\right) \odot \boldsymbol{I}\right)\boldsymbol{S} + \boldsymbol{S}\left(\left(\boldsymbol{S}\left(\boldsymbol{S}^2 \odot \boldsymbol{I}\right)\boldsymbol{S}^2\right) \odot \boldsymbol{I}\right) \\
& - 8\left(\boldsymbol{S}^2 \odot \boldsymbol{I}\right)\left(\boldsymbol{S}^2 \odot \boldsymbol{S}\right) - 8\left(\boldsymbol{S}^2 \odot \boldsymbol{S}\right)\left(\boldsymbol{S}^2 \odot \boldsymbol{I}\right) \\
& + 2\boldsymbol{S}\left(\boldsymbol{S}^2 \odot \boldsymbol{S}^2 \odot \boldsymbol{I}\right)\boldsymbol{S} + \left(\boldsymbol{S}^2 \odot \boldsymbol{S}^2 \odot \boldsymbol{S}\right)\boldsymbol{S} + \boldsymbol{S}\left(\boldsymbol{S}^2 \odot \boldsymbol{S}^2 \odot \boldsymbol{S}\right) \\
& - 3\left(\boldsymbol{S}^2 \odot \boldsymbol{S}^2\right) - 3\boldsymbol{S} \odot \left(\boldsymbol{S}\left(\boldsymbol{S}^2 \odot \boldsymbol{S}\right)\right) \\
& - 3\boldsymbol{S} \odot \left(\left(\boldsymbol{S}^2 \odot \boldsymbol{S}\right)\boldsymbol{S}\right) - 2\boldsymbol{S} \odot \left(\boldsymbol{S}\left(\boldsymbol{S}^2 \odot \boldsymbol{I}\right)\boldsymbol{S}\right) - 3\boldsymbol{S} \odot \left(\boldsymbol{S}\left(\boldsymbol{S} \odot \boldsymbol{S}^2\right)\right) \quad (124)
\end{aligned}
$$

$$
\begin{aligned}
\boldsymbol{P}_6^3 = & - 3\boldsymbol{S} \odot \left(\left(\boldsymbol{S} \odot \boldsymbol{S}^2\right)\boldsymbol{S}\right) + \boldsymbol{S}\left(\left(\boldsymbol{S}\left(\boldsymbol{S}^2 \odot \boldsymbol{I}\right)\boldsymbol{S}\right) \odot \boldsymbol{I}\right)\boldsymbol{S} \\
& - 3\left(\boldsymbol{S} \odot \boldsymbol{S}^2\right)\boldsymbol{S} - 3\boldsymbol{S}\left(\boldsymbol{S} \odot \boldsymbol{S}^2\right) + \boldsymbol{S} \odot \boldsymbol{S}^4 + 4\boldsymbol{S}^4 \\
& + \left(\boldsymbol{S} \odot \boldsymbol{S}^3\right)\boldsymbol{S} + \boldsymbol{S}\left(\boldsymbol{S} \odot \boldsymbol{S}^3\right) - 4\left(\boldsymbol{I} \odot \boldsymbol{S}^3\right)\boldsymbol{S} - 4\boldsymbol{S}\left(\boldsymbol{I} \odot \boldsymbol{S}^3\right) \\
& + \left(\boldsymbol{S} \odot \boldsymbol{S}^2\right)\boldsymbol{S}^2 + \boldsymbol{S}^2\left(\boldsymbol{S} \odot \boldsymbol{S}^2\right) - 4\left(\boldsymbol{I} \odot \boldsymbol{S}^2\right)\left(\boldsymbol{S} \odot \boldsymbol{S}^2\right) - 4\left(\boldsymbol{S} \odot \boldsymbol{S}^2\right)\left(\boldsymbol{I} \odot \boldsymbol{S}^2\right) \\
& + 3\boldsymbol{S}\left(\boldsymbol{S} \odot \boldsymbol{S}^2\right)\boldsymbol{S} - 4\left(\left(\boldsymbol{S}\left(\boldsymbol{S}^2 \odot \boldsymbol{S}\right)\right) \odot \boldsymbol{I}\right)\boldsymbol{S} - 4\boldsymbol{S}\left(\left(\boldsymbol{S}\left(\boldsymbol{S}^2 \odot \boldsymbol{S}\right)\right) \odot \boldsymbol{I}\right) \\
& - 2\boldsymbol{S} \odot \left(\boldsymbol{S}\left(\boldsymbol{S} \odot \boldsymbol{S}^2\right)\right) - 2\boldsymbol{S} \odot \left(\left(\boldsymbol{S} \odot \boldsymbol{S}^2\right)\boldsymbol{S}\right) \quad (125)
\end{aligned}
$$

$$
\begin{aligned}
\boldsymbol{P}_6^4 = & - 4\left(\boldsymbol{S}^2 \odot \boldsymbol{I}\right)\boldsymbol{S}^2 - 4\boldsymbol{S}^2\left(\boldsymbol{S}^2 \odot \boldsymbol{I}\right) - 8\boldsymbol{S}\left(\boldsymbol{S}^2 \odot \boldsymbol{I}\right)\boldsymbol{S} - \boldsymbol{S}^2\left(\boldsymbol{S}^2 \odot \boldsymbol{I}\right) - \left(\boldsymbol{S}^2 \odot \boldsymbol{I}\right)\boldsymbol{S}^2 \\
& - 4\left(\left(\left(\boldsymbol{S}^2 \odot \boldsymbol{S}\right)\boldsymbol{S}\right) \odot \boldsymbol{I}\right)\boldsymbol{S} - 4\boldsymbol{S}\left(\left(\boldsymbol{S}\left(\boldsymbol{S}^2 \odot \boldsymbol{S}\right)\right) \odot \boldsymbol{I}\right) - \boldsymbol{S} \odot \left(\boldsymbol{S}\left(\boldsymbol{S}^2 \odot \boldsymbol{I}\right)\boldsymbol{S}\right) \\
& - \left(\boldsymbol{S}^3 \odot \boldsymbol{I}\right)\boldsymbol{S} - \boldsymbol{S}\left(\boldsymbol{S}^3 \odot \boldsymbol{I}\right) - 2\left(\boldsymbol{S}^2 \odot \boldsymbol{I}\right)\boldsymbol{S}^2 - 2\boldsymbol{S}^2\left(\boldsymbol{S}^2 \odot \boldsymbol{I}\right) \\
& - \boldsymbol{S}^2 \odot \boldsymbol{S}^2 - 2\boldsymbol{S}\left(\boldsymbol{S}^2 \odot \boldsymbol{I}\right)\boldsymbol{S} \\
& - \left(\boldsymbol{S}^2 \odot \boldsymbol{S}\right)\boldsymbol{S} - \boldsymbol{S}\left(\boldsymbol{S}^2 \odot \boldsymbol{S}\right) + 44\boldsymbol{S}^2 \odot \boldsymbol{S} + 12\boldsymbol{S}^2 \quad (126)
\end{aligned}
$$

### K.3 NODE LEVEL CYCLES FOR $k \leq 6$

For $k \leq 6$ the number of node level cycles can be computed as $\frac{1}{2}\left(\boldsymbol{P}_k \odot \boldsymbol{I}\right)\boldsymbol{1}$. The following expressions are derived after some algebraic manipulations.

$$
\left(\boldsymbol{P}_3 \odot \boldsymbol{I}\right)\boldsymbol{1} = \left(\boldsymbol{S}^3 \odot \boldsymbol{I}\right)\boldsymbol{1} = \left(\boldsymbol{S}^2 \odot \boldsymbol{S}\right)\boldsymbol{1} \quad (127)
$$

$$(P_4 \odot I)\, 1 = (S^2 \odot S^2)\, 1 - 2 (S^2 \odot I)\, S1 - S (S^2 \odot I)\, 1 + 2S1 \tag{128}$$

$$(P_5 \odot I)\, 1 = (S^5 \odot I)\, 1 - 4 (S^3 \odot I)\, S1 - S (S^3 \odot I)\, 1 - 2 (S^2 \odot S)\, S1$$
$$+ 6 (S^2 \odot S)\, 1 - 4 (S \odot S^2 \odot I)\, 1 + 3 (S^3 \odot I)\, 1 \tag{129}$$

The terms involved in the expressions for $(P_3 \odot I)\, 1,\; (P_4 \odot I)\, 1,\; (P_5 \odot I)\, 1$ can all be computed by a GNN defined by 1 or 26, as shown in Appendix D.

$$(P_6^1 \odot I)\, 1 = (S^6 \odot I)\, 1 - 2 (S^4 \odot I)\, S1$$
$$- 2 (S^2 \odot S^4 \odot I)\, 1 - S (S^4 \odot I)\, 1$$
$$- 2 (S^3 \odot S^3 \odot I)\, 1 - 2 (S^3 \odot S)\, S1$$
$$- 2 (S^2 \odot S)(S^3 \odot I)\, 1$$
$$+ 4 (S^3 \odot S)\, 1$$
$$- (S^2 \odot S^2)\, S1 + 6 (S^2 \odot S^2 \odot S^2 \odot I)\, 1 \tag{130}$$

$$(P_6^2 \odot I)\, 1 = 4 (S^2 \odot S^2 \odot S)\, 1$$
$$+ 4 (S^2 \odot I)\, S^2 1$$
$$+ 2S (S^2 \odot S^2 \odot I)\, 1 + 6 (S^2 \odot S^2 \odot S)\, 1$$
$$- 3 (S^2 \odot S^2 \odot I)\, 1, \tag{131}$$

where we used the following expression.

$$\mathrm{diag}((S^2 \odot I)\, S (S^2 \odot I)\, S) = (S^2 \odot I)\, S^2 1 \tag{132}$$

The terms involved in the $(P_6^1 \odot I)\, 1,\; (P_6^2 \odot I)\, 1$ can be computed by a GNN defined by 1 or 26, as shown in Appendix D.

$$(P_6^3 \odot I)\, 1 = S^3 1 - 6 (S \odot S^2)\, 1 + 6 (S^2 \odot S^2)\, 1 + 2 (S^2 \odot S \odot S^2)\, 1$$
$$+ 3\mathrm{diag}(S (S \odot S^2)\, S) \tag{133}$$

The last term can be computed by a two-layer GNN defined by 26, where $\sigma (\cdot) = (\cdot)^k$, $\rho$ is equal to identity after the first layer and $\rho [\cdot] = \mathbb{E} [\cdot]$ after the second layer. Thorough analysis is presented in Appendix E.

$$(P_6^4 \odot I)\, 1 = -15 (S^2 \odot I)\, S1 - 10S^2 1 - 2 (S^2 \odot S)\, 1 + 12S1 \tag{134}$$

The terms involved in the $(P_6^3 \odot I)\, 1,\; (P_6^4 \odot I)\, 1$ can be computed by a GNN defined by 1 or 26, as shown in Appendix D, and Appendix E.

### K.4 Node level cycles for $k = 7$

For $k = 7$ the number of node level cycles can be computed as $\frac{1}{2} (P_6 \odot S)\, 1$ or $\frac{1}{2}\mathrm{diag} (SP_6)$. The following expressions are derived after some algebraic manipulations.

$$
\begin{aligned}
\left(\boldsymbol{P}_6^1 \odot \boldsymbol{S}\right) \mathbf{1} = &\left(\boldsymbol{S}^4 \odot \boldsymbol{S}^3\right) \mathbf{1} \\
&- 2\left(\boldsymbol{S}^5 \odot \boldsymbol{S}^2 \odot \boldsymbol{I}\right) \mathbf{1} - \boldsymbol{S}\left(\boldsymbol{S}^3 \odot \boldsymbol{S}^2\right) \mathbf{1} - \boldsymbol{S}\left(\boldsymbol{S}^4 \odot \boldsymbol{S}\right) \boldsymbol{S} \mathbf{1} \\
&- 2\left(\boldsymbol{S}^2 \odot \boldsymbol{S}\right)\left(\boldsymbol{S}^4 \odot \boldsymbol{I}\right) \mathbf{1} - 2\left(\boldsymbol{S}^4 \odot \boldsymbol{S}^3 \odot \boldsymbol{I}\right) \mathbf{1} + \left(\boldsymbol{S}^4 \odot \boldsymbol{S}\right) \mathbf{1} \\
&- 2\left(\boldsymbol{S}^3 \odot \boldsymbol{S}\right)\left(\boldsymbol{S}^3 \odot \boldsymbol{I}\right) \mathbf{1} - 2\left(\boldsymbol{S}^3 \odot \boldsymbol{S}^2\right) \boldsymbol{S} \mathbf{1} - \left(\boldsymbol{S}^4 \odot \boldsymbol{S}\right) \boldsymbol{S} \mathbf{1} \\
&- \left(\boldsymbol{S}^2 \odot \boldsymbol{S}^2\right)\left(\boldsymbol{S}^3 \odot \boldsymbol{I}\right) \mathbf{1} \\
&+ 4\boldsymbol{S}\left(\boldsymbol{S}^3 \odot \boldsymbol{S}^2 \odot \boldsymbol{I}\right) \mathbf{1} + 6\left(\boldsymbol{S}^3 \odot \boldsymbol{S}^2 \odot \boldsymbol{S}^2 \odot \boldsymbol{I}\right) \mathbf{1} + 8\left(\boldsymbol{S}^3 \odot \boldsymbol{S}^2 \odot \boldsymbol{S}\right) \mathbf{1} \\
&+ \operatorname{diag}\left(\boldsymbol{S}\left(\boldsymbol{S}^2 \odot \boldsymbol{I}\right) \boldsymbol{S}\left(\boldsymbol{S}^3 \odot \boldsymbol{I}\right)\right) + \left(\boldsymbol{S}^3 \odot \boldsymbol{I}\right) \boldsymbol{S}^2 \mathbf{1} \\
&+ 2 \operatorname{diag}\left(\boldsymbol{S}\left(\boldsymbol{S}^3 \odot \boldsymbol{S}\right) \boldsymbol{S}\right) \\
&+ \boldsymbol{S}^2\left(\boldsymbol{S}^3 \odot \boldsymbol{I}\right) \mathbf{1} + \left(\boldsymbol{S}^2 \odot \boldsymbol{I}\right) \boldsymbol{S}\left(\boldsymbol{S}^3 \odot \boldsymbol{I}\right) \mathbf{1} \\
&+ 2\left(\boldsymbol{S}^2 \odot \boldsymbol{S}\right)\left(\boldsymbol{S}^2 \odot \boldsymbol{S}^2 \odot \boldsymbol{I}\right) \mathbf{1} \\
&+ \left(\boldsymbol{S}^2 \odot \boldsymbol{S}^2 \odot \boldsymbol{S}^2 \odot \boldsymbol{S}\right) \mathbf{1} + \operatorname{diag}\left(\boldsymbol{S}\left(\boldsymbol{S}^2 \odot \boldsymbol{I}\right) \boldsymbol{S}^2\left(\boldsymbol{S}^2 \odot \boldsymbol{I}\right)\right),
\end{aligned} \tag{135}
$$

where we have used the following identities:

$$
\left(\boldsymbol{S}^2 \odot \boldsymbol{I}\right) \boldsymbol{S}\left(\boldsymbol{S}^3 \odot \boldsymbol{I}\right) \mathbf{1}
$$

$$
\boldsymbol{S}\left(\boldsymbol{S}^3 \odot \boldsymbol{I}\right) \boldsymbol{S}\left(\boldsymbol{S}^2 \odot \boldsymbol{I}\right) = \boldsymbol{S}\left(\boldsymbol{S} \odot \operatorname{vec}\left(\boldsymbol{S}^3 \odot \boldsymbol{I}\right) \operatorname{vec}\left(\boldsymbol{S}^2 \odot \boldsymbol{I}\right)^T\right)
$$

$$
\begin{aligned}
\operatorname{diag}\left(\boldsymbol{S}\left(\boldsymbol{S}^3 \odot \boldsymbol{I}\right) \boldsymbol{S}\left(\boldsymbol{S}^2 \odot \boldsymbol{I}\right)\right) &= \left(\boldsymbol{S} \odot \operatorname{vec}\left(\boldsymbol{S}^3 \odot \boldsymbol{I}\right) \operatorname{vec}\left(\boldsymbol{S}^2 \odot \boldsymbol{I}\right)^T\right) \mathbf{1} \\
&= \left(\boldsymbol{S}^3 \odot \boldsymbol{I}\right) \boldsymbol{S}\left(\boldsymbol{S}^2 \odot \boldsymbol{I}\right) \mathbf{1} = \left(\boldsymbol{S}^3 \odot \boldsymbol{I}\right) \boldsymbol{S}^2 \mathbf{1}
\end{aligned}
$$

The terms involved in the $\left(\boldsymbol{P}_6^1 \odot \boldsymbol{S}\right) \mathbf{1}$ can be computed by a GNN defined by 1 or 26, as shown in Appendix D, and Appendix E.

$$
\begin{aligned}
\left(\boldsymbol{P}_6^2 \odot \boldsymbol{S}\right) \mathbf{1} = &2 \operatorname{diag}\left(\boldsymbol{S}\left(\boldsymbol{S}^2 \odot \boldsymbol{S}\right) \boldsymbol{S}^2\right) + 2\left(\boldsymbol{S}^3 \odot \boldsymbol{S}^2 \odot \boldsymbol{S}\right) \mathbf{1} \\
&+ \left(\boldsymbol{S}^2 \odot \boldsymbol{S}\right) \boldsymbol{S}^2 \mathbf{1} + \left(\boldsymbol{S}^3 \odot \boldsymbol{I}\right) \boldsymbol{S}^2 \mathbf{1} \\
&+ \operatorname{diag}\left(\boldsymbol{S}\left(\boldsymbol{S}^2 \odot \boldsymbol{I}\right) \boldsymbol{S}\left(\boldsymbol{S}^2 \odot \boldsymbol{I}\right) \boldsymbol{S}\right) + \operatorname{diag}\left(\boldsymbol{S}^2\left(\boldsymbol{S}^2 \odot \boldsymbol{I}\right) \boldsymbol{S}\left(\boldsymbol{S}^2 \odot \boldsymbol{I}\right)\right) \\
&+ \boldsymbol{S}\left(\boldsymbol{S}^2 \odot \boldsymbol{S}\right) \boldsymbol{S} \mathbf{1} \\
&+ 2 \operatorname{diag}\left(\boldsymbol{S}\left(\boldsymbol{S} \odot \boldsymbol{S}^2 \odot \boldsymbol{S}^2\right) \boldsymbol{S}\right) + 2\left(\boldsymbol{S}^2 \odot \boldsymbol{S}^2 \odot \boldsymbol{S}^2 \odot \boldsymbol{S}\right) \mathbf{1} \\
&+ \boldsymbol{S}\left(\boldsymbol{S}^2 \odot \boldsymbol{S}\right) \boldsymbol{S} \mathbf{1} + 2\left(\boldsymbol{S}^2 \odot \boldsymbol{I}\right)\left(\boldsymbol{S}^2 \odot \boldsymbol{S}\right)\left(\boldsymbol{S}^2 \odot \boldsymbol{I}\right) \mathbf{1} \\
&- 8\left(\boldsymbol{S}^2 \odot \boldsymbol{S}\right) \boldsymbol{S} \mathbf{1} - 8 \operatorname{diag}\left(\boldsymbol{S}\left(\boldsymbol{S}^2 \odot \boldsymbol{S}\right)\left(\boldsymbol{S}^2 \odot \boldsymbol{I}\right)\right) \\
&+ 2\left(\boldsymbol{S}^2 \odot \boldsymbol{S}\right)\left(\boldsymbol{S}^2 \odot \boldsymbol{S}^2 \odot \boldsymbol{I}\right) \mathbf{1} + \boldsymbol{S}\left(\boldsymbol{S}^2 \odot \boldsymbol{S}^2 \odot \boldsymbol{S}\right) \mathbf{1} + \left(\boldsymbol{S}^2 \odot \boldsymbol{S}^2 \odot \boldsymbol{S}^2 \odot \boldsymbol{S}\right) \mathbf{1} \\
&- 3\left(\boldsymbol{S}^2 \odot \boldsymbol{S}^2 \odot \boldsymbol{S}\right) \mathbf{1} \\
&- 6 \operatorname{diag}\left(\boldsymbol{S}\left(\boldsymbol{S}^2 \odot \boldsymbol{S}\right) \boldsymbol{S}\right) - 3\left(\boldsymbol{S}^2 \odot \boldsymbol{S}^2 \odot \boldsymbol{S}\right) \mathbf{1} \\
&- 2\left(\boldsymbol{S}^2 \odot \boldsymbol{S}\right) \boldsymbol{S} \mathbf{1}
\end{aligned} \tag{136}
$$

$$
\begin{aligned}
\left(\boldsymbol{P}_6^3 \odot \boldsymbol{S}\right) \mathbf{1} = &-\left(\boldsymbol{S}^2 \odot \boldsymbol{S}^2 \odot \boldsymbol{S}\right) \mathbf{1} + \left(\boldsymbol{S}^2 \odot \boldsymbol{S}\right) \boldsymbol{S}^2 \mathbf{1} \\
&- 5 \operatorname{diag}\left(\boldsymbol{S}\left(\boldsymbol{S} \odot \boldsymbol{S}^2\right) \boldsymbol{S}\right) - 5\left(\boldsymbol{S}^2 \odot \boldsymbol{S}^2 \odot \boldsymbol{S}\right) \mathbf{1} + 5\left(\boldsymbol{S}^3 \odot \boldsymbol{S}^2\right) \mathbf{1} \\
&+ \operatorname{diag}\left(\boldsymbol{S}\left(\boldsymbol{S} \odot \boldsymbol{S}^3\right) \boldsymbol{S}\right) - 4\boldsymbol{S}\left(\boldsymbol{S}^3 \odot \boldsymbol{I}\right) \mathbf{1} - 4\left(\boldsymbol{S}^3 \odot \boldsymbol{S}^2 \odot \boldsymbol{I}\right) \mathbf{1} \\
&+ 4 \operatorname{diag}\left(\boldsymbol{S}\left(\boldsymbol{S} \odot \boldsymbol{S}^2\right) \boldsymbol{S}^2\right) - 8\left(\boldsymbol{S}^2 \odot \boldsymbol{S}\right) \boldsymbol{S} \mathbf{1} \\
&- 4\boldsymbol{S}\left(\boldsymbol{S}^2 \odot \boldsymbol{S}\right) \mathbf{1} - 4\left(\boldsymbol{S}^2 \odot \boldsymbol{I}\right)\left(\boldsymbol{S}^2 \odot \boldsymbol{S}\right) \mathbf{1}
\end{aligned} \tag{137}
$$

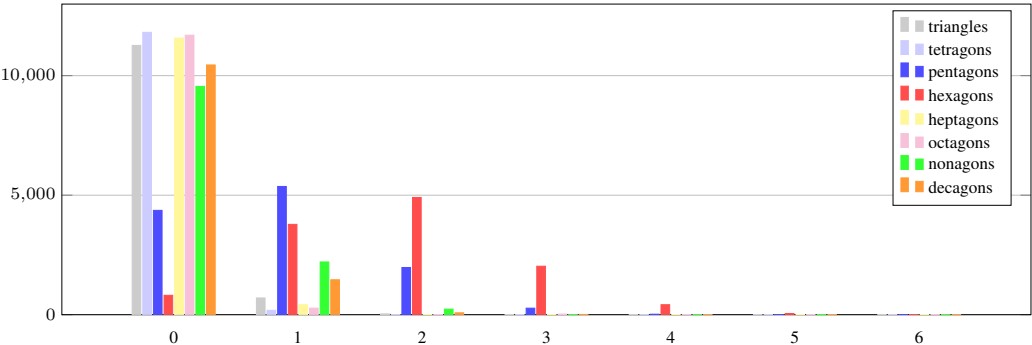

Figure 2: Histogram of cycles that appear in the ZINC dataset. The $x$ axis indicated the number of cycles in each graph and the $y$ axis shows the number of times each of the cycle counts appears.

$$\left(\boldsymbol{P}_6^4 \odot \boldsymbol{S}\right) \mathbf{1} = -18 \left(\boldsymbol{S}^2 \odot \boldsymbol{S}\right) \boldsymbol{S} \mathbf{1} - 7 \left(\boldsymbol{S}^3 \odot \boldsymbol{I}\right) \boldsymbol{S} \mathbf{1}$$
$$-5 \boldsymbol{S} \left(\boldsymbol{S}^2 \odot \boldsymbol{S}\right) \mathbf{1} - 4 \left(\boldsymbol{S}^2 \odot \boldsymbol{I}\right) \left(\boldsymbol{S}^2 \odot \boldsymbol{S}\right) \mathbf{1}$$
$$- \boldsymbol{S} \left(\boldsymbol{S}^3 \odot \boldsymbol{I}\right) \mathbf{1} - \left(\boldsymbol{S}^2 \odot \boldsymbol{I}\right) \left(\boldsymbol{S}^3 \odot \boldsymbol{I}\right) \mathbf{1}$$
$$- \left(\boldsymbol{S}^2 \odot \boldsymbol{S}^2 \odot \boldsymbol{S}\right) \mathbf{1}$$
$$- \left(\boldsymbol{S}^2 \odot \boldsymbol{S}^2 \odot \boldsymbol{S}\right) \mathbf{1} + 56 \left(\boldsymbol{S}^2 \odot \boldsymbol{S}\right) \mathbf{1} \tag{138}$$

All the terms involved in the $\left(\boldsymbol{P}_6 \odot \boldsymbol{S}\right) \mathbf{1}$ can be computed by a GNN defined by 1 or 26, as shown in Appendix D, and Appendix E.

## L COMPUTATIONAL AND MEMORY COMPLEXITY

There are two different ways to implement the proposed `Moment-GNN`. The first is to feed it with random vectors and compute the empirical moments, and the second is to work directly with the equivalent model. In the first scenario, the complexity of `Moment-GNN` is exactly the same as the complexity of a message-passing GNN. For instance, the memory complexity of the forward pass of GIN is $\Theta\left(|\mathcal{V}|F\right)$, where $F$ is the number of graph filters (hidden dimension, or width) in each layer, and the computational complexity is $\Theta\left(|\mathcal{V}|F^2 + |\mathcal{E}|F\right)$. If we use the equivalent model. We need to perform the following type of operations, *before training or testing*.

$$\left(\boldsymbol{S}^{i_1} \odot \cdots \odot \boldsymbol{S}^{i_k}\right) \left(\boldsymbol{S}^{i_{k+1}} \odot \cdots \odot \boldsymbol{S}^{i_m}\right) \mathbf{1}$$

As a result, the forward pass memory, and computational complexity of `Moment-GNN` is $\Theta\left(|\mathcal{V}|F\right)$, and $\Theta\left(|\mathcal{V}|F^2 + |\mathcal{E}|F\right)$ respectively. However, there is an additional preprocessing memory complexity of $\Theta\left(\text{nnz}\left(\boldsymbol{S}^{max(i_k)}\right)\right)$ and computational complexity of $\Theta\left(|\mathcal{V}||\mathcal{E}|\right)$, nnz counts the number of nonzeros in a matrix. Note that, if the sparsity of the matrices is very high specialized sparse matrix multiplication algorithms can be employed to further reduce the computational complexity. Additionally, since $max\left(i_k\right)$ is usually small (in our proofs it is less than or equal to 5) the memory complexity is in the order of $|\mathcal{E}|$.

## M EXPERIMENTS

In this section, we discuss the implementation details of the experiments presented in Section 7. We also present baseline comparisons in detecting the presence of $9-$node and $10-$node cycles in the ZINC dataset. The statistics of ZINC are illustrated in Fig. 2.

Table 6: Classification accuracy for cycle detection in ZINC

| Algorithm | Nonagon | | Decagon | |
|---|---|---|---|---|
| | Training | Testing | Training | Testing |
| `Moment-GNN` | **100** | **99.8** | **99.9** | **99.4** |
| `SMP` | **100** | 99.6 | 99.8 | 98.2 |
| `GIN + rand id` | 92.4 | 89.8 | 88.5 | 77.0 |
| `Ring-GNN` | 98.3 | 98.3 | 88.5 | 87.9 |
| `PPGN` | 93.9 | 91.1 | 88.0 | 87.1 |

## M.1 IMPLEMENTATION DETAILS

The experiments are conducted on a Linux server with NVIDIA RTX 3080 GPU. The code of the proposed architecture with all the experiments can be found in this repository[3].

To perform cycle detection (graph classification) we use the specifications in `https://github.com/cvignac/SMP` In particular, the `Moment-GNN` layer is followed by a 6-layer GIN with, which we train with stochastic gradient descent optimizer, initial learning rate equal to $10^{-3}$, batch size equal to 16 and a dropout ratio equal to 0.5. For the synthetic data experiment, we use 300 epochs to train whereas for the ZINC data experiment, we use 1000 epochs. The hidden dimension for the GNN layers is 32 and for the output (classification layer) 128.

To perform cycle counting (graph regression) we use the specifications in `https://github.com/gbouritsas/GSN`. The `Moment-GNN` layer is followed by 2 MPNN layers, which are trained with Adam for 1000 epochs, initial learning rate equal to $10^{-2}$, and batch size equal to 16. Each layer is followed by a 3-layer MLP with ReLU activation function and a batch normalization layer. The hidden dimension for the GNN layers is 128. In this set of experiments, we use CPU implementation.

## M.2 BASELINE COMPARISONS

Following the discussion in Section 7.3, we compare the performance of the proposed `Moment-GNN` in the last two tasks, i.e., detect a nonagon or a decagon in the ZINC dataset. As mentioned in Section 7.3, we train and test with a subset of the available graphs to ensure balanced classes. We use 70% of the graphs for training 20% for testing and 10% for validation. The baselines used for comparison are GIN with random input `GIN + rand id` Xu et al. (2019); Abboud et al. (2021); Sato et al. (2021), `PPGN` (Maron et al., 2019), `Ring-GNN` Chen et al. (2019), and `SMP` (Vignac et al., 2020).

The training and testing results for detecting nonagons and decagons are presented in Table 6. We observe that detecting a decagon is a more challenging task in this dataset. `Moment-GNN` achieves the best testing performance, whereas `SMP` demonstrates a similar performance. Compared to the remaining baselines, `Moment-GNN` is at least 11.5% percent better in detecting decagons, 1.5% better than `Ring-GNN` and at least 8% better than `GIN + rand id` and `PPGN` in detecting nonagons. It is notable that although both `Moment-GNN` and `GIN + rand id` use random input, `Moment-GNN` is markedly better in both tasks. This is due to the fact that our proposed framework is able to maintain the pivotal property of permutation equivariance, contrary to `GIN + rand id`.

In logP prediction and graph classification we use the following baselines:

**logP prediction Baselines:** `GCN` Kipf & Welling (2016), `GIN` Xu et al. (2019), `GraphSage` Hamilton et al. (2017), `GAT` Veličković et al. (2018), `MoNet` Gilmer et al. (2017), `GatedGCN` Bresson & Laurent (2017), **GIN + rand id** `PNA` Corso et al. (2020), `DGN` Beaini et al. (2021), `GNNML` Balcilar et al. (2021), `HIMP` Fey et al. (2020), `SMP` Vignac et al. (2020), and `GSN with cycles` Bouritsas et al. (2022), `CIN` Bodnar et al. (2021), `CIN-small` Bodnar et al. (2021), `GraphSage+sub` Barceló et al. (2021), `GAT+sub` Barceló et al. (2021), `MoNet+sub` Barceló et al. (2021), `GatedGCN+sub` Barceló et al. (2021), `GCN+sub` Barceló et al. (2021).

---

[3]https://github.com/MomentGNN/Counting

Table 7: logP Prediction in ZINC

| GNN Model | MAE | MAE (EF) |
|---|---|---|
| GraphSage | $0.410 \pm 0.005$ | − |
| GraphSage+sub | $0.24 \pm 0.01$ | − |
| GAT | $0.463 \pm 0.002$ | − |
| GAT+sub | $0.22 \pm 0.010$ | − |
| MoNet | $0.407 \pm 0.007$ | − |
| MoNet+sub | $0.16 \pm 0.01$ | − |
| GatedGCN | $0.422 \pm 0.006$ | $0.363 \pm 0.009$ |
| GatedGCN+sub | $\mathbf{0.135} \pm 0.01$ | − |
| PNA | $0.320 \pm 0.032$ | $0.188 \pm 0.004$ |
| DGN | $0.219 \pm 0.0102$ | $0.168 \pm 0.003$ |
| GNNML | $0.161 \pm 0.006$ | − |
| HIMP | − | $0.151 \pm 0.006$ |
| SMP | $0.219\pm$ | $0.138\pm$ |
| GIN + rand id | $0.322 \pm 0.026$ | $0.279 \pm 0.023$ |
| GCN | $0.469 \pm 0.002$ | − |
| GCN+sub | $0.20 \pm 0.01$ | − |
| GIN | $0.254 \pm 0.014$ | $0.209 \pm 0.018$ |
| GSN with cycles | $\mathbf{0.140 \pm 0.006}$ | $0.115 \pm 0.012$ |
| **Moment-GNN** | $\mathbf{0.140 \pm 0.004}$ | $\mathbf{0.110 \pm 0.005}$ |

Table 8: logP Prediction in ZINC-full

| Method | MAE |
|---|---|
| HIMP | $0.036 \pm 0.002$ |
| CIN-small | $0.044 \pm 0.003$ |
| GIN | $0.088 \pm 0.002$ |
| **Moment-GNN** | $0.041 \pm 0.001$ |

**Graph Classification Baselines:** GCN Kipf & Welling (2016), GIN Xu et al. (2019), Bresson & Laurent (2017), GIN + rand id, GSN with cliques Bouritsas et al. (2022), AWL Ivanov & Burnaev (2018), DGK Yanardag & Vishwanathan (2015), PATCHYSAN Niepert et al. (2016) RetGK Zhang et al. (2018), WLkernel Shervashidze et al. (2011), and WEGL Kolouri et al. (2020).

We also compare against some extra baselines in Barceló et al. (2021). The new comparisons can be found in Table 7.

## M.3 ZINC-FULL

We also test the performance of the proposed Moment-GNN in the task of logP prediction using the full version of the ZINC dataset. ZINC-full contains about 250,000 molecular graphs with up to 38 heavy atoms. Without adjusting any hyperparameter of the architecture used for the small ZINC dataset that contains 12,000 graphs, we are able to achieve MAE between the true and predicted logP value in the order of $10^{-2}$. We compare against GIN and HIMP which are generic architectures as Moment-GNN, and CIN which is a specific architecture for molecular graph processing. The results are reported in Table 8. Note that in the table we report the small version of CIN that uses a similar number of parameters as our architecture for fair comparisons. The full version of CIN achieves MAE $0.022 \pm 0.002$.

