# OpenReview forum: "Counting Graph Substructures with Graph Neural Networks"
_ICLR.cc/2024/Conference — ICLR 2024 poster_

### Official Review · Reviewer_QJcy · 2023-10-30

**Soundness:** 2 fair
**Presentation:** 1 poor
**Contribution:** 2 fair
**Rating:** 6
**Confidence:** 3

**Summary:**

This paper focuses on the representation power of GNNs and their ability to count graph substructures. The analysis enables the design of a generic GNN architecture that achieves good performance in various tasks, including subgraph detection, subgraph counting, graph classification, and logP prediction. The paper also provides both theoretical and experimental evidence of the ability of GNN to count graph substructures in graph-level representation learning tasks.

**Strengths:**

1. The paper studies an important problem: the ability to recognize subgraphs is crucial in graph applications.

2. The authors provide detailed deductions of how to design the proposed GNN to identify different numbers of graph substructures.

**Weaknesses:**

1. Writing quality: The presentation and the layout of the paper need to be improved substantially. Equation spacing/referencing is not correct with editing hints unremoved.
The paper is hard to follow. A clear introduction/navigation at the beginning of each section is lacking. This is particularly the case in the methodology part, which confuses the reader.

2. Experiments: The experiments show that the proposed GNN has good capability of substructure counting. However, this may not lead to better performance on downstream tasks. There are many benchmark datasets commonly used for node-level and graph-level tasks. Testing on more datasets will better demonstrate how counting substructures can help improve the performance of other tasks.

3. For the regression task on ZINC, the method proposed in ‘Graph Neural Networks with Local Graph Parameters’ accepted by NeurIPS 2021 seems to have a better result, which is not included in this paper.

**Questions:**

See the weaknesses above.

---

> ### Author Response · Authors · 2023-11-19
> **Response to Reviewer's comments**
>
> > Writing quality: The presentation and the layout of the paper need to be improved substantially. Equation spacing/referencing is not correct with editing hints unremoved. The paper is hard to follow. A clear introduction/navigation at the beginning of each section is lacking. This is particularly the case in the methodology part, which confuses the reader.
>
> We would like to thank the reviewer for their suggestion. We have substantially improved the paper presentation in the revised manuscript. We would be happy to implement further changes that the reviewer may suggest after reviewing our revised manuscript.
>
> > Experiments: The experiments show that the proposed GNN has good capability of substructure counting. However, this may not lead to better performance on downstream tasks. The model was tested on two datasets only, which is not sufficient. There are many benchmark datasets commonly used for node-level and graph-level tasks. Testing on more datasets will better demonstrate how counting substructures can help improve the performance of other tasks.
>
> We would like to thank the reviewer for their comment. We want to point out that we are testing our approach in 5 different datasets. Two synthetically generated datasets, to test the main scope of the paper (counting substructures), and 3 real benchmark datasets (ZINC, REDDIT-B, REDDIT-M) to assess the performance and generality of our approach in downstream tasks. We are also adding experiments with the full ZINC dataset in the revised version and would be happy to include additional datasets that the reviewer suggests. However, we believe that the current set of experiments is sufficient to show the effectiveness and generality of our approach. Note that the contribution of our work is not only practical but has a strong analytical component which we prefer to maintain.
>
> > The proposed model relies on the outcomes from other subgraph recognition studies as the ground truth for training. This limits its applicability as its performance may be affected by these upstream models.
>
> We would like to respectfully correct the reviewer in this statement. Our proposed model is completely independent of other subgraph recognition studies. We do not use any subgraph information from other models during training, validation, or testing.
>
> > For the regression task on ZINC, the method proposed in ‘Graph Neural Networks with Local Graph Parameters’ accepted by NeurIPS 2021 seems to have a better result, which is not included in this paper.
>
> Thank you for bringing this work to our attention, which we cite and include in the revised version of the paper. However, it is not true that this method achieves better performance. In fact, the best performance for the regression task on ZINC, which is reported in ‘Graph Neural Networks with Local Graph Parameters’ is MAE 0.1353 ± 0.01, whereas our proposed approach achieves MAE 0.110 ± 0.005.

---

> > ### Comment · Reviewer_QJcy · 2023-11-22
> > **Thank you for the response**
> >
> > The rebuttal has addressed my concerns. I appreciate the authors' effort in including the full ZINC dataset in evaluation and improving the writing clarity in the revision. I have raised my rating.

---

> > > ### Author Response · Authors · 2023-11-22
> > > **Thank you for your feedback**
> > >
> > > Thank you for taking the time to review our paper. Your input was valuable and we are glad to address your concerns.

---

> ### Author Response · Authors · 2023-11-22
> **We would like to hear your thoughts on our response and revised manuscript**
>
> Dear reviewer,
>
> We very much appreciate your feedback and suggestions. Your input was important in improving our paper. We believe that the writing of our manuscript has been significantly improved. We have also added one more dataset and comparisons with ‘Graph Neural Networks with Local Graph Parameters’, as requested. We also clarified that our approach does not depend on other subgraph methods.
>
> We would really appreciate it if you could find time to share your thoughts on our response and the revised manuscript.

---

### Official Review · Reviewer_K7V4 · 2023-10-31

**Soundness:** 3 good
**Presentation:** 3 good
**Contribution:** 2 fair
**Rating:** 6
**Confidence:** 3

**Summary:**

This work studies the substructure counting power of a message-passing neural networks (MPNN) with random input. Concretely, the function class of interest is the expectation of outputs of standard MPNNs with random node features. They prove that this model can count up to 7 cycles, which is more powerful than regular MPNN with constant input. Moreover, the model is shown be equivalent to a deterministic graph filtering with constant input, where the filter is a hadamard product of polynomial filters. The equivalent filtering model has a better empirical performance over tasks such as substructure counting and molecular property prediction.

**Strengths:**

- Though it is known that random node feature injection can improve expressive power, it is interesting to see such analysis in an average meaning (thus perserving equivariance), and the connection to the corresponding graph filtering.

**Weaknesses:**

- The comparison to other baselines on cycle counting and ZINC is insufficient.
- I feel like the sentence in the abstract "However, their ability to count substructures, which play a crucial role in biological and social networks, remains uncertain" may be confusing in the sense that, the counting ability of regular MPNN is pretty clear as shown in [1].


[1] Zhengdao Chen, Lei Chen, Soledad Villar, and Joan Bruna. Can graph neural networks count substructures? Advances in neural information processing systems, 33:10383–10395, 2020.

**Questions:**

- I was wondering if large cycles (such as 8 cycles) can also be written as hadamard product of $S^k$ and thus can be expressed by the model? I know the construction of such function is pretty combinatorial, so probably the question is open. But it is interesting as 7 cycle is also the counting limitation of 2-FWL.

---

> ### Author Response · Authors · 2023-11-19
> **Response to reviewer's comments**
>
> > The comparison to other baselines on cycle counting and ZINC is insufficient.
>
> Note that our approach is generic and we try to keep the comparisons between more general approaches for fairness. Nevertheless, we have compared the performance of our approach against 25 different baselines in our experiments. We are also including 5 more baselines from ‘Graph Neural Networks with Local Graph Parameters’ on ZINC in the revised manuscript. However, we would be happy to include any further comparisons/discussions that the reviewer may suggest.
>
>
> > I feel like the sentence in the abstract "However, their ability to count substructures, which play a crucial role in biological and social networks, remains uncertain" may be confusing in the sense that, the counting ability of regular MPNN is pretty clear as shown in [1].
>
> We agree with the reviewer that [1] has provided significant insights into the counting ability of message-passing GNNs. [1] studies the counting capabilities of message-passing GNNs with either constant or symmetric (structural) input signals. In that case, the counting capacity of message-passing GNNs is upper bounded by the 1-FWL test. However, our approach shows that random input signals can better exploit the power of message-passing operations and improve their counting capacity beyond 1-FWL. As a result, the ability of message-passing GNNs to count substructures is indeed very clear with symmetric input signals, thanks to [1], however when the input is not symmetric this research question is still open.
>
> To avoid possible confusion we rephrase this part to 'However, there are still open questions regarding their ability to count and list substructures, which play a crucial role in biological and social networks."
>
>
> > I was wondering if large cycles (such as 8 cycles) can also be written as hadamard product of $\mathbf{S}^k$ and thus can be expressed by the model? I know the construction of such function is pretty combinatorial, so probably the question is open. But it is interesting as 7 cycle is also the counting limitation of 2-FWL.
>
> In our new results, we can prove that 8 cycles can be written using Hadamard products of $\mathbf{S}^k$ and Khatri-Rao products of $\mathbf{S}^k$. These operations can actually be computed by message-passing GNNs with random inputs as discussed in Section 6. As a result, message-passing GNNs with random inputs can produce equivariant representations that are more powerful than the 2-FWL test for certain graph classes. Regarding the model in (14) (in the revised manuscript), we have not proven yet whether it is bounded by 2-FWL test or not. Please see the general response and Section 6 in the revised manuscript for more details.

---

> > ### Comment · Reviewer_K7V4 · 2023-11-20
> > **Thank you for your response**
> >
> > I appreciate authors' efforts of providing more counting ability analysis and experiments during rebuttal. I keep my original rate.

---

> > > ### Author Response · Authors · 2023-11-22
> > > **Thank you for your feedback**
> > >
> > > Thank you for appreciating our work and giving us constructive feedback.

---

### Official Review · Reviewer_4Tmr · 2023-10-31

**Soundness:** 3 good
**Presentation:** 3 good
**Contribution:** 3 good
**Rating:** 6
**Confidence:** 3

**Summary:**

This paper studies the representation power of GNNs in terms of counting graph substructures, including cycles, quasi-cliques, and connected components. The authors employ tools from tensor algebra and stochastic processes, and they demonstrate that with appropriate activation and normalization functions, GNNs can generate permutation equivariant node representations that capture statistical moments of the GNN output distribution.

**Strengths:**

1. The exploration of the graph substructure counting ability in the context of GNNs constitutes a novel contribution to this field. This particular property is closely tied to the representational power of GNNs and is critical for numerous real world graph modalities like social networks, making it a pertinent and valuable topic for discussion.

2. The theoretical analysis presented in the paper is extensive and commendable, shedding light on various aspects of the subject matter.

3. The empirical experiments conducted in the paper aligns well with theoretical findings and demonstrates the effectiveness of proposed method.

**Weaknesses:**

1. Many powerful, expressive, and well-adopted  GNN baselines are missed. For instance, GNN-AK[1], GrphGPS[2], CIN [3], ....

2. The technique of encoding substructures in graphs is widely used in GNNs to improve the expressiveness, such as EGO [1], GNN-AK [1] and NGNN [4].  I think the comparison with these substructure-based GNNs should be discussed.

[1] Lingxiao Zhao, Wei Jin, Leman Akoglu, and Neil Shah. From stars to subgraphs: Uplifting any GNN with local structure awareness. In International Conference on Learning Representations,2022.

[2] Rampášek, L., Galkin, M., Dwivedi, V. P., Luu, A. T., Wolf, G., & Beaini, D. (2022). Recipe for a general, powerful, scalable graph transformer. Advances in Neural Information Processing Systems, 35, 14501-14515.

[3] Bodnar, C., Frasca, F., Otter, N., Wang, Y., Lio, P., Montufar, G. F., & Bronstein, M. (2021). Weisfeiler and lehman go cellular: Cw networks. Advances in Neural Information Processing Systems, 34, 2625-2640.

[4] Zhang, M., & Li, P. (2021). Nested graph neural networks. Advances in Neural Information Processing Systems, 34, 15734-15747.

[5]  Beatrice Bevilacqua, Fabrizio Frasca, Derek Lim, Balasubramaniam Srinivasan, Chen Cai, Gopinath Balamurugan, Michael M Bronstein, and Haggai Maron. Equivariant subgraph aggregation networks. In International Conference on Learning Representations, 2022.

**Questions:**

1. When performing graph regression task on dataset ZINC, does the 10000 training graphs, 1000 validation graphs and 1000 test graphs are randomly sampled or they are from the ZINC-100K? For a more fair comparison, can this experiment implemented on whole ZINC dataset?

2. Proposition 5.2 show that Moment-GNN can improve expressivity over 1-FWL. What about the upper-bound on its' expressive power?  Is it bounded by 3-WL like subgraph GNNs [1]?

[1] Frasca, F., Bevilacqua, B., Bronstein, M., & Maron, H. (2022). Understanding and extending subgraph gnns by rethinking their symmetries. Advances in Neural Information Processing Systems, 35, 31376-31390.

---

> ### Author Response · Authors · 2023-11-19
> **Response to Reviewer's comments**
>
> > Many powerful, expressive, and well-adopted GNN baselines are missed. For instance, GNN-AK[1], GrphGPS[2], CIN [3], ....
>
> We would like to thank the reviewer for bringing these works to our attention. We have cited and added discussions on your suggested papers in the revised manuscript.
>
> > The technique of encoding substructures in graphs is widely used in GNNs to improve the expressiveness, such as EGO [1], GNN-AK [1] and NGNN [4]. I think the comparison with these substructure-based GNNs should be discussed.
>
> We agree with the reviewer that comparison with these substructure methods is important and is discussed in the revised manusript. The basic difference between our approach and [1], [4] is that we learn to process substructures without any prior information or additional algorithmic help, whereas [1], [4] predefine the subgraphs of interest and resort to non-message passing mechanisms to identify them. After identifying these subgraphs they perform subgraph message-passing to learn local representations that are used to extract global graph embeddings. Both approaches have pros and cons. For instance, our proposed method is more generic as it is purely data-driven. However, it does not utilize the help of other algorithms or prior knowledge as [1], [4] which can be valuable in many tasks.
>
> > When performing graph regression task on dataset ZINC, does the 10000 training graphs, 1000 validation graphs and 1000 test graphs are randomly sampled or they are from the ZINC-100K? For a more fair comparison, can this experiment implemented on whole ZINC dataset?
>
> They are not randomly sampled. The dataset and the training/testing/validation splits are downloaded from the Benchmarking Graph Neural Networks repository (https://github.com/graphdeeplearning/benchmarking-gnns). Thank you for your suggestion. We are currently testing our method on the whole ZINC dataset, which is also downloaded from the same repository. Without fine tuning our current test MAE is $0.040\pm 0.002$ among 6 different runs.
>
> > Proposition 5.2 show that Moment-GNN can improve expressivity over 1-FWL. What about the upper-bound on its' expressive power? Is it bounded by 3-WL like subgraph GNNs [1]?
>
> We would like to thank the reviewer for motivating us to look deeper into the expressivity of our approach. In our new results, we can prove that message-passing GNNs with random inputs can count 8-node cycles and 4-node cliques. These operations however do not admit a closed-form expression with purely Hadamard product operations but they also involve diagonals of the tensor Tucker model. As a result, the expressivity of message-passing GNNs with random inputs and the expectation operator in a hidden layer is not bounded by the 3-WL test. However, it could be the case that Moment-GNN (as defined in (14) in the revised manuscript) is indeed bounded by the 3-WL test, but we haven't proven it yet. Please see the general response and Section 6 in the revised manuscript for more details.

---

> ### Author Response · Authors · 2023-11-22
> **We would like to hear your thoughts on our response and revised manuscript**
>
> Dear reviewer,
>
> We would like to express our gratitude for your appreciation of our work. Your comments have been instrumental in deriving new results and enhancing our paper. We believe we have addressed all of your comments and would be delighted to hear your thoughts on our response and revised manuscript.

---

> > ### Comment · Reviewer_4Tmr · 2023-11-22
> > **Thank you for these responses**
> >
> > Thank you for your comprehensive response, and I have no further questions.

---

> ### Author Response · Authors · 2023-11-22
> **Thank you for your review**
>
> Thank you for appreciating our work and providing valuable comments. We are glad to answer all of your concerns.

---

### Official Review · Reviewer_jgwt · 2023-11-09

**Soundness:** 3 good
**Presentation:** 1 poor
**Contribution:** 2 fair
**Rating:** 6
**Confidence:** 3

**Summary:**

This paper studied the problem of detecting/counting graph structures such as cycles, cliques, and connected components, using MPNNs with random initial node features. They considered taking the high-order moment of the neural network output to obtain deterministic node representations. In this case, the authors proved that with the increase of the order, the resulting GNN can express more and more graph structures, resulting in higher expressivity. Experiments demonstrate the expressive power of the proposed method.

**Strengths:**

1. The theoretical results of this paper are rigorous and correct (although proving it is straightforward given prior work).
2. The proposed method is quite interesting. Prior to this work, researchers mainly improved the expressive power of GNNs by sacrificing the computational costs, e.g., using higher-order GNNs. While in this paper, the proposed GNNs only have linear complexity. On the other hand, using random node initialization has been proposed in prior work. However, unlike prior work, this paper achieves equivariance by using an expectation in the final layer (although there may be other weaknesses, see below). The authors showed promising theoretical results for this architectural design.

**Weaknesses:**

1. This paper is poorly written. Please carefully polish the paper in the rebuttal period. Several problems include: (1) the word "equation" is redundant in many places; (2) many definitions are unclear. For example, what do you mean by "x is anonymous" in page 4, and what do you mean by "stationary random vector"? (3) the paper even exposes the author name "Charilaos" in page 6. Please fix it. (4) What is role of the characteristic function in page 3? Why is it unrelated to x?

2. Regarding the theoretical results:
   - The authors only proved positive results for the expressive power of GNNs using high-order moments, which are incomplete. Does the GNNs fail without high-order moments? In other words, are the theoretical results tight? For example, can the GNNs count 6-cycle using only 2-order moment, and can the GNNs count 7-cycle using only 3-order moment?
   - I do not think Theorem 4.1 is meaningful. The theorem uses a GNN to fit the number of connected components of a *single* graph, which is just equivalent to fit a constant if I understand correctly.

3. Regarding the experiments:
   - The proposed method relies crucially on taking the expectation in the final layer. How do you implement this in your experiments? Will a large number of samples be needed?
   - I found from the results that the GNNs can even count 8-cycles perfectly but the current theory did not prove it. What is the number of moment order required in your experiments?

   Please give more details in your paper.

**Questions:**

See the box above.

---

> ### Author Response · Authors · 2023-11-19
> **Response to Reviewer's comments (1/2)**
>
> > The theoretical results of this paper are rigorous and correct (although proving it is straightforward given prior work).
>
> We want to thank the reviewer for appreciating our work. It is worth mentioning that our proofs are quite complex, involving the analysis of multiple layers of GNNs with nonlinearities in between (see Section 5 and Appendix F). To the best of our knowledge, this is one of the few attempts that analyze multiple GNN layers.
>
> > This paper is poorly written. Please carefully polish the paper in the rebuttal period. Several problems include: (1) the word "equation" is redundant in many places;
>
> Thank you for your suggestion. We have significantly improved the paper writing in the revised version. Unfortunately the word "equation" was generated automatically from the ICLR template whenever the command \eqref was used, which explains the repetition. We have addressed this issue in the revised version.
>
> > many definitions are unclear. For example, what do you mean by "x is anonymous" in page 4, and what do you mean by "stationary random vector"?
>
> By "anonymous x," we refer to node features that are completely uninformative, i.e., they possess no information about the nodes or the graph structure. In other words, they are agnostic to the graph and the node identities. In the revised manuscript, we changed "anonymous" to "uninformative." The term "stationary random vector" is well-defined in probability theory and refers to the characteristics of the distribution of the input. We have removed this term in the revised manuscript to avoid confusion.
>
> Please see our answer to the next comment for further explanations.
>
> > What is role of the characteristic function in page 3? Why is it unrelated to x?
>
> The short answer is that input x is a random vector which can be fully described by its characteristic function, the same way a random vector can be fully described by its density function (e.g., Gaussian function). The characteristic function can also be viewed as the Fourier Transform of the density function. Since x can be completely described by its characteristic function it is also completely related to x.
>
> We understand that there is a confusion regarding input x and its properties and therefore have updated the first part of Section 3, which now reads:
>
> The goal of our paper is to answer the research question in 1.1 and characterize the expressive power of message-passing GNNs in terms of counting important substructures of the graph. To do that we study  Equation 2, when the input is a random vector $\mathbf{x}\in\mathbb{R}^N$. Our work focuses on the ability of message-passing GNN operations to generate expressive features, therefore the input $\mathbf{x}$ should be completely uninformative, i.e., structure and identity agnostic. To ensure that $\mathbf{x}=\left[x_1,x_2,\dots,x_N\right]^T$ is agnostic with respect to the structure of the graph, we assume that $x_i$, $i=1,\dots,N$ are independent random variables. To ensure that $x_i$s are identity agnostic, we assume that they are identically distributed and satisfy $\mathbb{E}\left[ x_i\right]= 0$ $\mathbb{E}\left[ x_i^p\right]= 1$, for $i=1,\dots,N$, and $p\geq 2\in\mathbb{Z}$. Overall, the elements of $\mathbf{x}$ are independent and identically distributed (i.i.d.), with joint characteristic function $\phi_{\mathbf{x}}\left(t_1,\dots,t_N\right)=\prod_{i=1}^N \left(e^{jt_i}-jt_i\right)$, and the moments of the joint distribution of $\mathbf{x}$ satisfy:
>
> $\mathbb{E}\left[\mathbf{x}\right]=\mathbf{0},\quad
>     \mathbb{E}\left[\mathbf{x}\mathbf{x}^T\right]=\mathbf{I},\quad
>    \mathbb{E}\left[\mathbf{x}\circ\mathbf{x}\circ\mathbf{x}\right]=\underline{\mathbf{I}},\quad
>     \mathbb{E}\left[\mathbf{x}\circ\mathbf{x}\circ\dots\circ\mathbf{x}\right]=\underline{\mathbf{I}}$

---

> ### Author Response · Authors · 2023-11-19
> **Response to Reviewer's comments (2/2)**
>
> > The authors only proved positive results for the expressive power of GNNs using high-order moments, which are incomplete. Does the GNNs fail without high-order moments? In other words, are the theoretical results tight? For example, can the GNNs count 6-cycle using only 2-order moment, and can the GNNs count 7-cycle using only 3-order moment?
>
> The point the reviewer is raising is important and is worth further clarification. Our results are sufficient but not necessary. For example, we cannot guarantee that 7-node cycles cannot be computed using the 3rd-order moments. However, it is important to note that the question we are addressing in this paper is not whether high-order moments of certain graph filters can count substructures, but whether message-passing GNNs can count substructures. High-order moments are not the goal but rather the tool we are using to analyze the counting abilities of GNNs.
>
> For instance, a 2-layer GNN with square nonlinearities implicitly or explicitly computes 4th-order moments, and a 3-layer GNN with square nonlinearities computes 6th-order moments. To make things more interesting, GNNs with analytic pointwise nonlinearities, such as the hyperbolic tangent, can linearly combine infinitely high-order information in their output with just a single layer. As a result, the question the reviewer is posing is indeed very interesting, but more from a graph theory perspective, rather than GNN analysis.
>
> > I do not think Theorem 4.1 is meaningful. The theorem uses a GNN to fit the number of connected components of a single graph, which is just equivalent to fit a constant if I understand correctly.
>
> We respectfully disagree with the reviewer's comment. Theorem 4.1 demonstrates that message-passing GNN operations are able to generate powerful node embeddings that, when summed across nodes, enable the computation of the number of connected components in a graph. Considering that message-passing GNNs with constant or structural inputs are provably unable to generate this information, we find the significance of Theorem 4.1 noteworthy.
>
> However, we agree with the reviewer that our other theorems prove more powerful results. In particular, we establish that message-passing GNNs can learn how to count cycles and quasi-cliques for any graph. As a result, our other theorems not only characterize the expressive power of GNNs but also their generalization ability.
>
>
> In summary, Theorem 4.1 is crucial in describing the expressive power of GNNs and is applicable to graphs observed during training. On the other hand, all the other theorems characterize both the expressive and generalization power of GNNs and are also applicable to graphs that were not observed during training. Parts of this discussion appear in the revised manuscript.
>
> > The proposed method relies crucially on taking the expectation in the final layer. How do you implement this in your experiments? Will a large number of samples be needed?
>
> In fact the proposed approach **does not** necessarily rely on taking the expectation in the final layer. One can either implement the architecture with the expectation or use the equivalent closed-form expression of Equation 14. In our implementation, we employ the equivalent closed-form expression. Since we are using a single Moment GNN layer followed by GIN layers, the exact expressions are described in Equation 65 of the Appendix. Detailed discussion about this topic can be found in Proposition 5.1, Remark 5.1, and Section 7.1 of the revised manuscript.
>
> > I found from the results that the GNNs can even count 8-cycles perfectly but the current theory did not prove it. What is the number of moment order required in your experiments?
>
> Please see the general comment.
>
> Our new results, which are presented in Section 6 of the revised manuscript, prove that message-passing operations can actually count 8-node cycles. These results however do not directly apply to the implementation we used in our experiments. In the experiments, the moment order is a hyperparameter which is selected between 3-5.

---

> > ### Comment · Reviewer_jgwt · 2023-11-22
> > **Thank you**
> >
> > Thank you for your thorough reply. Several of my questions have been addressed. However, I am still not convinced by the significance of Theorem 4.1. Since there is only one graph, fitting the connected component should be trivial. Can you explain why you said that "Considering that message-passing GNNs with constant or structural inputs are provably unable to generate this information"?
> >
> > Another weakness is in the implementation. You said that you use the equivalent closed-form expression of Equation 14 instead of taking expectation. However, this is quite computationally expensive (unlike the standard MPNN). It requires at least $\Omega(n^3)$ computational costs. It is not clear what is the advantage of using your GNN design in practice.
> >
> > Finally, it seems that you have added many new results in the updated paper. I should acknowledge that **I may not have enough time to carefully check all the new results again**, but after a coarse check it seems that the writing quality has been improved.

---

> > > ### Author Response · Authors · 2023-11-22
> > > **Thank you for your answer. There may be a misunderstanding, we kindly ask you to reconsider.**
> > >
> > > Thank you for acknowledging that several of your questions have been addressed and taking the time to respond. We kindly ask you to read our response to your updated questions, since we believe there might be a misunderstanding.
> > >
> > > >However, I am still not convinced by the significance of Theorem 4.1. Since there is only one graph, fitting the connected component should be trivial.
> > >
> > > As we mentioned in the previous comment, the implications of Theorem 4.1 **does not just fit a single graph to one constant, but learns how to count the connected components of all graphs that are observed during training**. To avoid any future confusion we have adjusted the Theorem that now reads:
> > >
> > > **Theorem 4.1** [Number of connected components]
> > >
> > > *Given a set of graphs $\{\mathcal{G}_i\}$, $i=1$,$\dots$,$M$, there exists a GNN defined in 1 or 2 that counts the number of connected components of all $\{\mathcal{G}_i\}$, $i=1$,$\dots$,$M$.*
> > >
> > > As you can see in Appendix G, **the proof is essentially the same for the previous and current version of Theorem 4.1**, and this is why the theorem was initially stated with one graph.
> > >
> > >
> > > >Can you explain why you said that "Considering that message-passing GNNs with constant or structural inputs are provably unable to generate this information"?
> > >
> > > Message-passing GNNs with constant or structural (symmetric) inputs are provably less powerful than the WL test (1-WL or 2-WL), thanks to the well-known works of Xu et al. (2019); Morris et al. (2019).
> > >
> > > The work of Chen et al. (2020) (https://arxiv.org/pdf/2002.04025.pdf) shows in their Theorem 3.3. that the 2-WL cannot induced-subgraph-count any connected pattern with 3 or more nodes, and therefore message-passing GNNs with constant or structural (symmetric) inputs cannot count the connected components **of any graph**.
> > >
> > >
> > > >Another weakness is in the implementation. You said that you use the equivalent closed-form expression of Equation 14 instead of taking expectation. However, this is quite computationally expensive (unlike the standard MPNN). It requires at least $\Omega(n^3)$ computational costs. It is not clear what is the advantage of using your GNN design in practice.
> > >
> > > It is not true that our computational complexity is at least $\Omega(n^3)$, since Equation 14 involves **sparse operations**. The computational complexity of our work is analyzed in Appendix M of both the revised and original manuscript. As we mention in Appendix M the extra computational complexity of Moment-GNN is ${\Theta}\left(|\mathcal{V}||\mathcal{E}|\right)$ to compute the adjacency powers. The Hadamard product operations are even sparser in practice, e.g., $\mathbf{S}^2\odot \mathbf{S}$ is always more sparse than $\mathbf{S}$.
> > >
> > > Testament to the low complexity of our work is the fact that we can handle the REDDIT-B and REDDIT-M datasets that methods with $\Omega(n^3)$ complexity cannot.
> > >
> > > Given our responses, we kindly ask the reviewer to reconsider and not judge our research based on a misunderstanding.

---

> ### Author Response · Authors · 2023-11-22
> **We would like to hear your thoughts on our response and revised manuscript**
>
> Dear reviewer,
>
> We would like to thank you for your comments and constructive feedback. Your review helped our paper to improve significantly.
>
> The writing of the revised manuscript has been significantly improved and the definitions are now super clear. We also believe that we have addressed your concerns regarding our theoretical results. Regarding experiments, we also clarified how our model is implemented in practice without the need to compute expectations.
>
> As a result, we would be very glad if you could share your thoughts on our response and revised manuscript. We would be happy to answer any further questions you may have.

---

> ### Comment · Reviewer_jgwt · 2023-11-22
> **Thank you for the prompt reply.**
>
> I believe both of my concerns on Theorem 4.1 and practical implementation are addressed. I tend towards raising my score. As a minor comment, please link each Appendix section to some place in the main text.

---

> > ### Author Response · Authors · 2023-11-22
> > **Thank you for all your comments**
> >
> > We would like to thank you for all your comments which significantly contributed to the improved quality of our paper. We implemented your suggestion regarding the Appendix in the newly revised manuscript.

---

### Author Response · Authors · 2023-11-19
**General comment by Authors**

We would like to thank the reviewers for their valuable comments and constructive feedback. The reviewers' comments helped us to significantly enhance the presentation, quality, and results of our paper. In the revised version of the paper, you can find the following main changes.

1. We include new results on the counting abilities of message-passing GNNs. In particular, we prove that by measuring high-order statistical moments, message-passing GNNs can learn to:


    a. **List** all the triangles of any graph.


    b. Count **4-node cliques** of any graph.


    c. Count **8-node cycles** of any graph.

  Our new results show that message-passing GNNs with random inputs and an expectation operator in a hidden layer can break the expressivity limitations of 2-FWL for certain classes of graphs. These results can be found in the newly formed Section 6 of the revised submission. To facilitate the discussion we also include Section 6 in this response.

2. We add additional experiments in the Appendix on the full ZINC dataset (per reviewer's 4Tmr suggestion), which contains about 250,000 molecular graphs with up to 38 heavy atoms. Without adjusting any hyperparameter of the architecture used for the small ZINC dataset that contains 12,000 graphs, we are able to achieve MAE between the true and predicted logP value equal to $0.040\pm 0.002$ among 2 different runs. Since this is a very large dataset, some results are still pending and will be reported before the rebuttal period ends.

3. We include additional comparisons with the methods in 'Graph Neural Networks with Local Graph Parameters’, per reviewer's QJcy and K7V4 suggestion. The results can be found in the Appendix. Our proposed Moment-GNN achieves better results in logP prediction on the ZINC dataset.


4. The paper presentation is substantially improved. In particular, we have edited Sections 3, 4, and 5 that present our proposed methodology, in a way that is clear and easy for the reader to follow. We have also added a small introduction at the beginning of each section per reviewer's QJcy suggestion. Equation spacing/referencing is also fixed with editing hints removed. We have also included further discussions with all the additional works that the reviewers suggested. The changes are marked in blue in the revised version.

---

> ### Author Response · Authors · 2023-11-19
> **Section 6 of revised manuscript: Listing and Counting structures beyond 2-FWL**
>
> In this section, we show how a message-passing GNN can generate equivariant features that list all the triangles of a graph, count the number of $4-$node cliques, $8-$node cycles, and therefore break the expressivity limits of the 2-FWL test. In particular, we consider again the GNN described in (2), which we study with an uninformative input as described in Section 3, to derive the following theorem.
>
> **Theorem [Listing Triangles]**
>
> There exists a message-passing GNN  defined below:
>
>   $  \underline{\mathbf{T}} = \text{ReLU}\left(\mathbb{E}\left[\mathbf{z}\circ\mathbf{z}\circ\mathbf{z}\right]\right)\in\{0,1\}^{N\times N\times N},\quad\mathbf{z} = \sum_{k=0}^{K}h_k{\mathbf{ S}}^k \mathbf{x},$
>
> with an uninformative input $\mathbf{x}\in\mathbb{R}^{N}$ as described in Section3 such that $ \underline{\mathbf{T}}[i,j,k] = 1$ if vertices $i,j,k$ form a triangle and $\underline{\mathbf{T}}[i,j,k]=0$ otherwise.
>
> In the above theorem we start with i.i.d. random node inputs,  as in the previous analysis, and perform linear message-passing operations to compute informative node representations $\mathbf{z}$. Instead of applying a pointwise nonlinearity to $\mathbf{z}$, we compute the three-way outer product $\mathbf{z}\circ\mathbf{z}\circ\mathbf{z}$, and process it via the expectation and ReLU. This enables the computation of tensor $\underline{\mathbf{T}}$, which encodes relations of node triplets and can list all the triangles in the graph. After learning tensor $\underline{\mathbf{T}}$ we can use it to generate a random vector $\mathbf{x}_{t}$ with the following statistical moments:
>
> $\mathbb{E}[\mathbf{x}_{t}]=\mathbf{0},$
>
> $\mathbb{E}[\mathbf{x}_{t}{\mathbf{x}_t}^T]=\mathbf{I},$
>
> $\mathbb{E}[{\mathbf{x}_t} \circ {\mathbf{x}_t} \circ {\mathbf{x}_t} ]=\underline{\mathbf{T}}$,
>
> $    \mathbb{E}[{\mathbf{x}_t} \circ\dots \circ {\mathbf{x}_t}]=\underline{\mathbf{I}},$
>
> We can then perform a similar analysis as the one in Section 3 and derive the following theorems.
>
> **Theorem[4-node cliques]**
>
> There exists a GNN defined in 1 or 2 with random input $\mathbf{x}_{t}$ defined above, that counts the number of 4-node cliques of any graph.
>
> **Theorem [8-node cycles]**
>
> There exists a GNN defined in 1 or 2 with random input $\mathbf{x}_{t}$ defined above, that counts the number of 8-node cycles of any graph.
>
> The details of the analysis can be found in the Appendix. The above theorems also prove that:
>
> **Proposition**
>
> There exist substructure families with $\mathcal{O}\left(1\right)$, such that 2-FWL is no stronger than the GNN described in the previous theorems.

---

### Meta-Review · Area_Chair_FoPT · 2023-12-06

**Metareview:**

This paper studies the substructure counting ability of a type of message-passing networks, where node features are randomly initialized and the output representation is obtained by taking expectation over the randomness. The authors proved that the architecture can count many common structures when using higher-order moments. Although this paper is poorly written in its initial submission, the authors have vastly revised the manuscript. Currently, this is a borderline paper where all reviewers lean towards acceptance.

**Justification For Why Not Higher Score:**

The experimental results are good but not strong enough.

**Justification For Why Not Lower Score:**

Reviewers are basically positive after the discussion.

---

### Decision · Program_Chairs · 2024-01-16

Accept (poster)